# Fibro-adipogenic progenitors of dystrophic mice are insensitive to NOTCH regulation of adipogenesis

Milica Marinkovic[1,*], Claudia Fuoco[1,*], Francesca Sacco[1,2], Andrea Cerquone Perpetuini[1], Giulio Giuliani[1], Elisa Micarelli[1], Theodora Pavlidou[1], Lucia Lisa Petrilli[1], Alessio Reggio[1], Federica Riccio[1], Filomena Spada[1], Simone Vumbaca[1], Alessandro Zuccotti[1], Luisa Castagnoli[1], Matthias Mann[2], Cesare Gargioli[1], Gianni Cesareni[1,3]

**Fibro-adipogenic progenitors (FAPs) promote satellite cell differentiation in adult skeletal muscle regeneration. However, in pathological conditions, FAPs are responsible for fibrosis and fatty infiltrations. Here we show that the NOTCH pathway negatively modulates FAP differentiation both in vitro and in vivo. However, FAPs isolated from young dystrophin-deficient *mdx* mice are insensitive to this control mechanism. An unbiased mass spectrometry–based proteomic analysis of FAPs from muscles of wild-type and *mdx* mice suggested that the synergistic cooperation between NOTCH and inflammatory signals controls FAP differentiation. Remarkably, we demonstrated that factors released by hematopoietic cells restore the sensitivity to NOTCH adipogenic inhibition in *mdx* FAPs. These results offer a basis for rationalizing pathological ectopic fat infiltrations in skeletal muscle and may suggest new therapeutic strategies to mitigate the detrimental effects of fat depositions in muscles of dystrophic patients.**

## Introduction

Skeletal muscle regeneration is a highly orchestrated process involving a variety of mononuclear cell populations that are either resident or attracted to the injured tissue by inflammatory signals (Bentzinger et al, 2013).

A stem cell population residing under the myofiber basal lamina, satellite cells (SCs), is the main source of myoblasts during regeneration (Wang & Rudnicki, 2012; Yin et al, 2013). As a consequence of the exhaustion of the SC stem cell pool in muscular dystrophy patients, the regeneration potential declines and excessive fibrosis and fat infiltrations take place (Chakkalakal et al, 2012; Rahimov & Kunkel, 2013). Intramuscular adipose tissue is one of the hallmarks of chronic myopathies, and its extent is a good indicator of disease progression, as it correlates with patient age and clinical stage (Gaeta et al, 2012).

A mesenchymal population of fibro-adipogenic progenitors (FAPs), which are located in the interstitial area of the skeletal muscle, positively regulates satellite activation and differentiation (Joe et al, 2010). During muscle regeneration caused by an acute insult, FAPs expand and promote myofiber repair by releasing paracrine factors that stimulate SC differentiation (Joe et al, 2010; Farup et al, 2015). Toward the end of the repair process, excessive FAPs, which are generated during the expansion phase, are removed while the remaining FAPs return to the initial quiescent state (Joe et al, 2010; Uezumi et al, 2010; Pretheeban et al, 2012; Lemos et al, 2015). In pathological conditions, instead of returning to the quiescent state, they rather differentiate causing fibrosis and fat infiltrations (Rodeheffer, 2010; Uezumi et al, 2010, 2011; Stumm et al, 2018).

The signals that regulate the choice between these alternative fates are still poorly characterized. When isolated from the muscle and cultivated ex vivo, FAPs differentiate spontaneously into adipocytes or fibroblasts. This implies that in vivo FAP differentiation is negatively controlled by signals from the muscle environment.

Non–cell-autonomous mechanisms mediated by factors synthetized by regenerating fibers play an important role in limiting adipogenesis during regeneration (Uezumi et al, 2010). Nitric oxide (NO) has also been reported to affect FAP adipogenic differentiation by down-regulation of the peroxisome proliferator-activated receptors gamma (PPARg) (Cordani et al, 2014). Along these lines, it has also been proposed that, in acutely damaged skeletal muscle, the balance between the levels of TNFa and TGFbsecreted by infiltrating inflammatory macrophages controls FAP function during regeneration (Lemos et al, 2015). In a mouse model (*mdx*) of Duchenne muscular dystrophy (DMD), the persistence of the anti-apoptotic TGFb signal prevents clearance of the amplified FAPs and favors their fibrotic differentiation. More recently, Kopinke and collaborators convincingly demonstrated a critical role of cilia in modulating the adipogenic fate of FAPs by controlling the activity of the Hedgehog pathway following muscle injury (Kopinke et al, 2017). It is not clear, however, if the shift of FAP fate toward adipogenic differentiation, occurring in myopathies, can be fully explained by

[1]Department of Biology, University of Rome Tor Vergata, Rome, Italy   [2]Department of Proteomics and Signal Transduction, Max-Planck Institute of Biochemistry, Martinsried, Germany   [3]Istituto di Ricovero e Cura a Carattere Scientifico (IRCCS) Fondazione Santa Lucia, Rome, Italy

Correspondence: cesareni@uniroma2.it; cesare.gargioli@uniroma2.it
*Milica Marinkovic and Claudia Fuoco contributed equally to this work.

an unbalance of these muscle environmental signals. In alternative, or in parallel, one should consider cell autonomous mechanisms whereby the mutant environment induces a metastable FAP state with a reduced sensitivity to anti-adipogenic signals.

Evolutionarily conserved pathways, such as Hedgehog, WNT, and NOTCH, have been implicated in the regulation of adipogenesis in a variety of experimental systems (Rosen & MacDougald, 2006). In the muscle system, in vitro studies suggest that a direct contact between myotubes and FAPs negatively affect FAP adipogenic differentiation (Uezumi et al, 2010; Huang et al, 2014). To investigate the molecular basis of this phenomenon, we focused on the NOTCH pathway because (i) it is activated by cell–cell contact (Andersson & Lendahl, 2014), (ii) it is a known regulator of stem cell quiescence (Koch et al, 2013), and (iii) it plays a fundamental role in muscle regeneration (Mourikis & Tajbakhsh, 2014).

Here we report that NOTCH also modulates FAP adipogenesis and that this control mechanism is compromised in FAPs from dystrophin-deficient, *mdx*, mice, a model of DMD. Our results support a model whereby the synergistic cooperation of NOTCH with other anti-adipogenic signals plays an important role in the regulation of FAP adipogenesis in both a healthy and a dystrophic muscle.

# Results

## High content single-cell analysis reveals that FAPs from regenerating muscles are phenotypically different

FAPs were purified from muscle mononuclear cells by the MACS microbead technology as CD31$^-$/CD45$^-$/$\alpha$7-integrin$^-$/SCA1$^+$ cells (from now on, FAPs) (Fig S1). In optimal conditions for adipogenic induction after 13 d in adipogenic induction medium (AIM), ~90% of the cells in this preparation differentiate into adipocytes (Uezumi et al, 2010). We first asked whether FAPs from *mdx* dystrophic mice can be phenotypically discriminated from wild type (*wt*). The *mdx* FAPs have different differentiation potentials in vivo and ex vivo when compared with *wt* FAPs (Mozzetta et al, 2013). We found that this phenotypic difference is reflected by differences in the surface protein expression profile as revealed by mass cytometry (Fig 1). For this analysis, we isolated and compared the antigen profiles of FAPs from 6-wk-old *wt* and *mdx* mice. At this age, the hind limb muscles of *mdx* mice are in a robust regeneration phase (Pastoret & Sebille, 1995). We also analyzed FAPs from a second model of muscle regeneration obtained by purifying mononuclear cells 3 d after cardiotoxin (*ctx*) injury of mice of comparable age (Fig 1A). Each condition (*wt*, *mdx*, and *ctx*) was analyzed in three biological replicates. The nine samples were each tagged with palladium isotopes, using a barcoding protocol (Bodenmiller et al, 2012). Cells were then combined in a single suspension and tagged with a heavy atom–labeled antibody panel including 12 antibodies (Figs 1 and S2A, and B) (Table S1). While most of the antigens showed a quasi-modal intensity distribution, with minor, albeit significant, shifts in average intensities (see for instance CD140b in Fig 1B), SCA1, CD34, and CD146 intensity distributions were clearly bimodal (Fig 1B). In addition to a low-expression peak characterizing the *wt* FAP

preparation, the *mdx* and *ctx* preparations displayed a second peak of cells expressing higher levels of CD34 and/or SCA1 (Fig 1B), the second population being more numerous in the *ctx* FAP preparations. The expression of SCA1 and CD34 were highly correlated, thus defining two FAP subpopulations with high or low expression of both antigens (circled in green and yellow in Fig 1C). The populations expressing anticorrelated levels of SCA1 and CD34 were of negligible size. The SCA1$^L$CD34$^L$ and SCA1$^H$CD34$^H$ subpopulations, expressing low and high levels of the two antigens, characterize the *wt* and *ctx* preparations, respectively, whereas the *mdx* FAP preparation contained both subpopulations with an approximately equal number of cells (Figs 1D and S2C).

The mass cytometry multiparametric data were analyzed using the viSNE algorithm (Amir el et al, 2013). By projecting the single-cell multiparametric data onto a two-dimensional plane, we could identify FAP subpopulations whose abundance changed in the different conditions. This analysis confirms that the CD31$^-$/CD45$^-$/$\alpha$7-integrin$^-$/SCA1$^+$ cells form a rather heterogeneous population whose states are influenced by the conditions of the muscle environment (Figs 1D and S2B).

In conclusion, high content single-cell analysis of the antigen expression profiles of FAPs from *wt, mdx* or *ctx* mice reveal remarkable and reproducible differences. The clearly recognizable viSNE two-dimensional plots suggest that the FAPs are in different cellular states depending on the muscle environment experienced in vivo. We speculated that these different states could underlie a different ability to respond to differentiation stimuli.

## The NOTCH signaling pathway modulates FAP differentiation into adipocytes in vitro

The single-cell analysis of FAPs from three experimental conditions (*wt, ctx* and *mdx* mice) detected a shift in the abundance of populations expressing different levels of FAP antigens. Notwithstanding these different antigen profiles, the three FAP preparations efficiently differentiate into adipocytes when isolated from the muscle environment and cultivated ex vivo (data not shown). We were interested in investigating whether the different cell states observed by mass cytometry analysis have the potential to affect, in a cell autonomous manner, their sensitivity to the inhibitory signals provided by other cell types in the muscle environment. We focused on the NOTCH pathway whose modulation had the most striking effect on FAP adipogenesis, without affecting cell number (Fig S3).

We isolated FAPs from *wt* mice and treated them with different concentrations of the $\gamma$-secretase inhibitor DAPT (Fig 2A) in growth medium (DMEM + 20% FBS) (Fig 2B). We observed a remarkable dose-dependent increase of the fraction of cells stained with Oil red O (ORO) when compared with untreated controls (Fig 2C and E). A similar trend was observed when adipogenic commitment was monitored by staining with antibodies against PPARg, the adipogenesis master regulator gene (Fig 2C and E). The average number of nuclei did not significantly change in treated groups compared with controls (Fig 2D). Because $\gamma$-secretase is essential for NOTCH activation, this observation is consistent with the hypothesis that NOTCH is involved in the regulation of adipogenesis.

We next asked whether exposure to the NOTCH ligand,Delta-like protein 1 (DLL1), affects FAP adipogenic differentiation. FAPs isolated

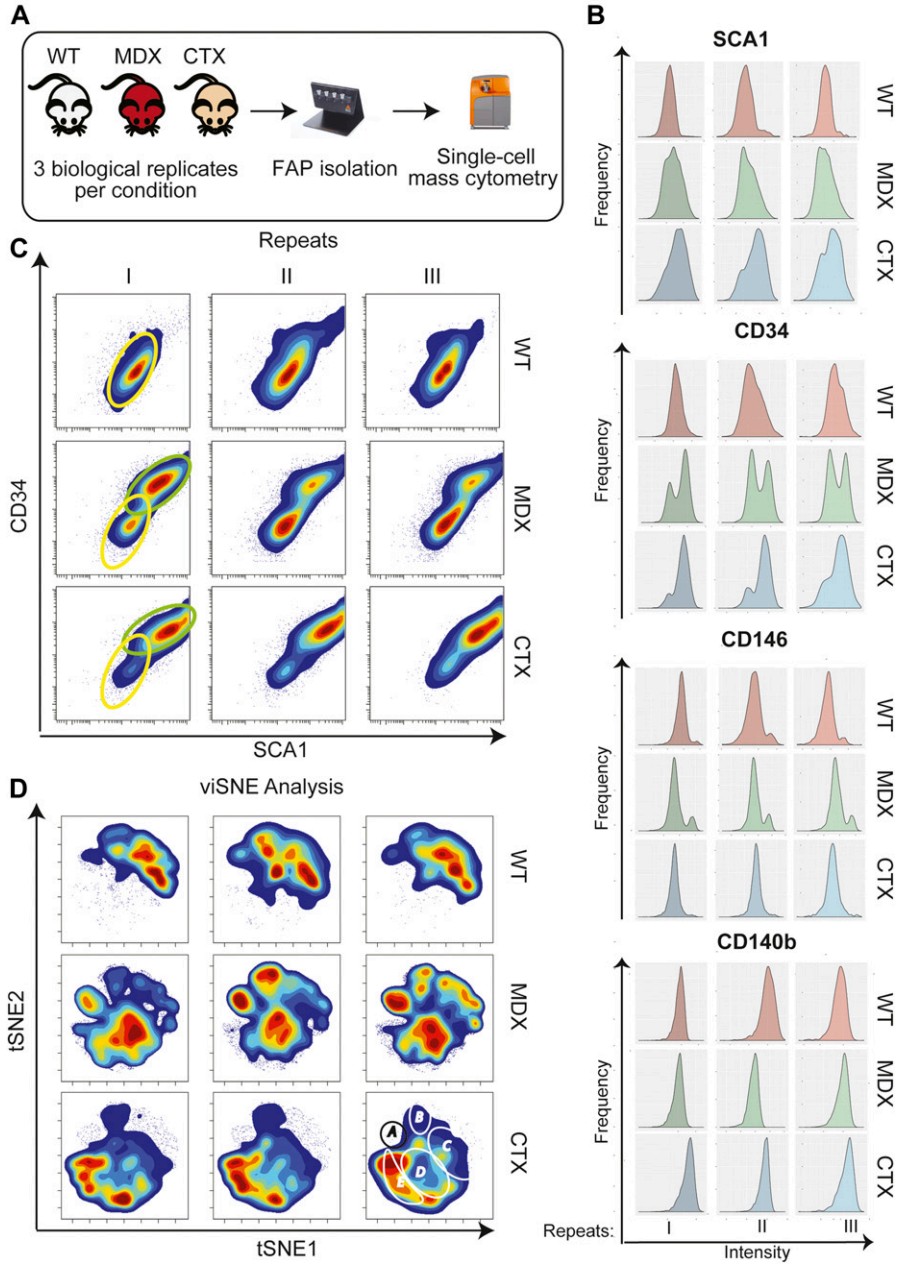

**Figure 1. Mass cytometry analysis of *wt*, *mdx*, and *ctx* FAPs.**

FAPs were purified using MACS microbead technology from hind limb muscles of 6-wk-old *wt*, *mdx*, and *ctx* (3 dpostinjury) mice. The different cell samples were bar coded by labeling with different proportions of three palladium isotopes. The bar coded cell samples were pooled and incubated with 12 metal-labeled antibodies (SCA1, CD34, CD146, CD140b, CD31, CXCR4, CAS3, Vim, pSTAT1, TNFa, pCREB and pSTAT3) and analyzed by mass cytometry in a Cytof2 instrument. **(A)** Schematic representation of the experimental design. **(B)** Intensity distribution of the signals of four different antigens (SCA1, CD34, CD146 and CD140b). **(C)** Contour maps of the scatter plots showing the correlations between SCA1 and CD34 expression in the three biological repeats for the three conditions. Colors represent cell densities: red = high, blue = low. The two different identified subpopulations are outlined by green (SCA1$^H$CD34$^H$) or yellow (SCA1$^L$CD34$^L$) ovals. **(D)** Contour representations of the viSNE maps in the nine different cell preparations. Regions of interest identifying cell subpopulations that characterize the different biological conditions are circled (A to E in the top left panel).

from *wt* mice were plated on a DLL1-Fc–coated surface and cultivated in growth medium for 8 d. One sample was additionally treated with 5 µM DAPT. FAPs seeded on plates coated with DLL1 showed a significant impairment of adipogenic differentiation. However, this inhibition was overridden by exposing cells to 5 µM DAPT (Fig 2F–H). The activation of NOTCH pathway and the inhibition of adipogenesis were monitored by Western blot analysis of HES-1 and PPARg expression, a downstream NOTCH target and the master transcription factor of adipogenesis, respectively (Fig 2H).

FAP cells, both wt and *mdx*, express NOTCH 1, 2 and 3 and to a lower degree NOTCH 4. FAPs also express NOTCH ligands (Figs S4 and S5).

To provide additional evidence of the involvement of NOTCH in FAP adipogenesis and to discriminate the contributions of the different NOTCH receptors, we knocked down the NOTCH receptors, using siRNA targeting *NOTCH 1, 2,* and *3* (Fig 3) on FAPs isolated from *wt* mice. The down-regulation of *NOTCH* transcripts was monitored by quantitative PCR (Fig 3D). Modulation of adipogenesis was evaluated by measuring PPARg expression after 48 h (Fig 3C) and the percentage of adipocytes after 8 d in culture (Fig 3A and B). Although knockdown of *NOTCH3* did not have much impact on adipogenic differentiation, the down-regulation of *NOTCH2* lead to a significant increase of FAP differentiation. Knockdown of *NOTCH1* slightly increased adipogenesis.

We conclude that FAPs, by synthesizing both NOTCH ligands and NOTCH receptors, are able to limit their own differentiation by an autocrine mechanism.

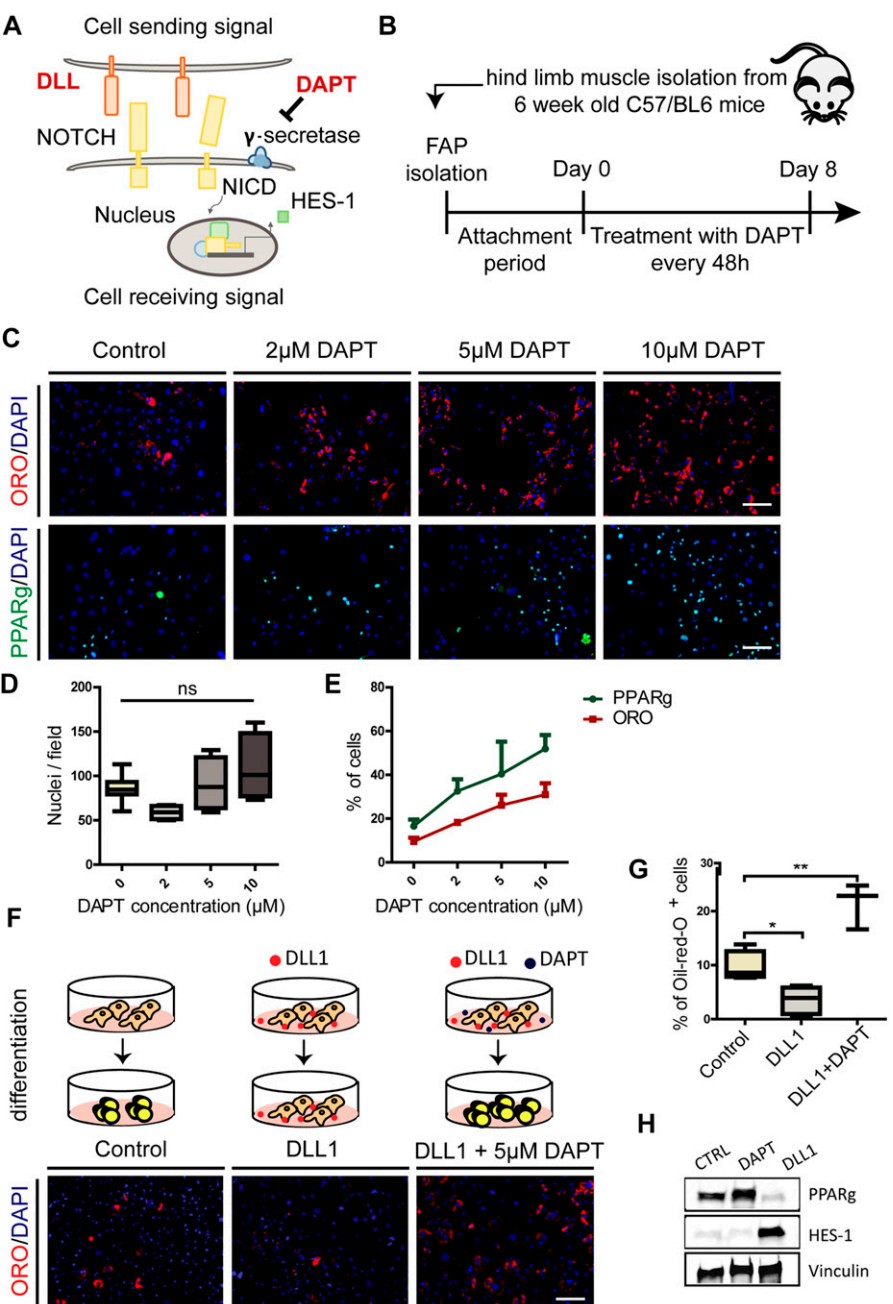

**Life Science Alliance**

**Figure 2. Perturbation of the NOTCH pathway modulates FAP differentiation into adipocytes.**
**(A)** Simplified model of canonical NOTCH signaling. DAPT (N-[N-(3,5-difluorophenacetyl)-ʟ-alanyl]-S-phenylglycine t-butylester) is a synthetic inhibitor of γ-secretase. **(B)** Schema of the experimental procedure. Attachment period lasted for 4 d. **(C)** FAPs were isolated from wild type (*wt*) C57BL/6 mice, cultivated in DMEM + 20% FBS, and treated upon attachment with 2, 5 and 10 μM of DAPT every 48 h for 8 days. Adipocytes were detected by Oil red O (ORO) stain or PPARg and nuclei were counterstained with DAPI. Scale bar: 100 μm. **(D)** The average number of nuclei are presented as a box plot (n = 4). **(E)** Line graph showing the percentage of ORO–positive and PPARg-positive cells represented as average ± SEM of four different experiments (n = 4). **(F)** FAPs isolated from hind limb muscles of C57BL/6 mice were plated on DLL1-Fc or IgG2A-Fc (control)–coated surface in DMEM + 20% FBS. In addition, FAPs were plated on DLL1-Fc–coated surface and treated with 5 μM DAPT every 48 h for 8 d. Adipocytes were detected with ORO stain and nuclei were counterstained with DAPI. Scale bar: 100 μm. **(G)** The percentage of ORO–positive cells of at least three different experiments are shown in the graphs. All quantifications were done using the CellProfiler software. Box plots show median and interquartile range with whiskers extended to minimum and maximum values. Statistical significance was evaluated by the ANOVA test (\*$P < 0.05$, \*\*$P < 0.01$). **(H)** Western blot analysis of HES-1 and PPARg protein levels in control, DAPT-treated and DLL1-treated cells. Proteins (30 μg) from FAP whole cell lysates were separated by electrophoresis on an SDS acrylamide (4–15%) gel and blotted onto nitrocellulose paper. Vinculin was used as a loading control (n = 3).

## Myotubes inhibition of FAP adipogenesis is NOTCH dependent

The cross talk between different cells in the skeletal muscle stem cell niche is crucial for the regulation of muscle homeostasis (Bentzinger et al, 2013). Myotubes can inhibit FAP differentiation when cocultured ex vivo (Uezumi et al, 2010; Huang et al, 2014). We confirmed that seeding purified muscle SCs and FAPs in coculture, in a 1:1 ratio, inhibits lipid droplets formation even in conditions favoring adipogenic differentiation (Fig 4B–D). To ensure adipogenic differentiation, cells were first stimulated with AIM for 48 h, which was then replaced with adipogenic maintenance medium for

four additional days (Fig 4A). As already reported by Uezumi et al (2010), we confirmed that adipogenic inhibition requires the direct contact between the two cell populations, as coculturing the two cell types in separate compartments, by using transwell inserts with 1-μm porous membrane, adipogenesis is not affected (Fig 4B–D). Adipocyte differentiation was estimated by monitoring the fraction of ORO–positive cells (Fig 4C) and by measuring the levels of adiponectin secreted in the medium (Fig 4D).

Given that in skeletal muscle the delta-like-ligand 1 (DLL1) is expressed on the cell membrane of SCs and myofibers (Conboy & Rando, 2002; Conboy et al, 2003; Kuang et al, 2007; Bi et al, 2016), and

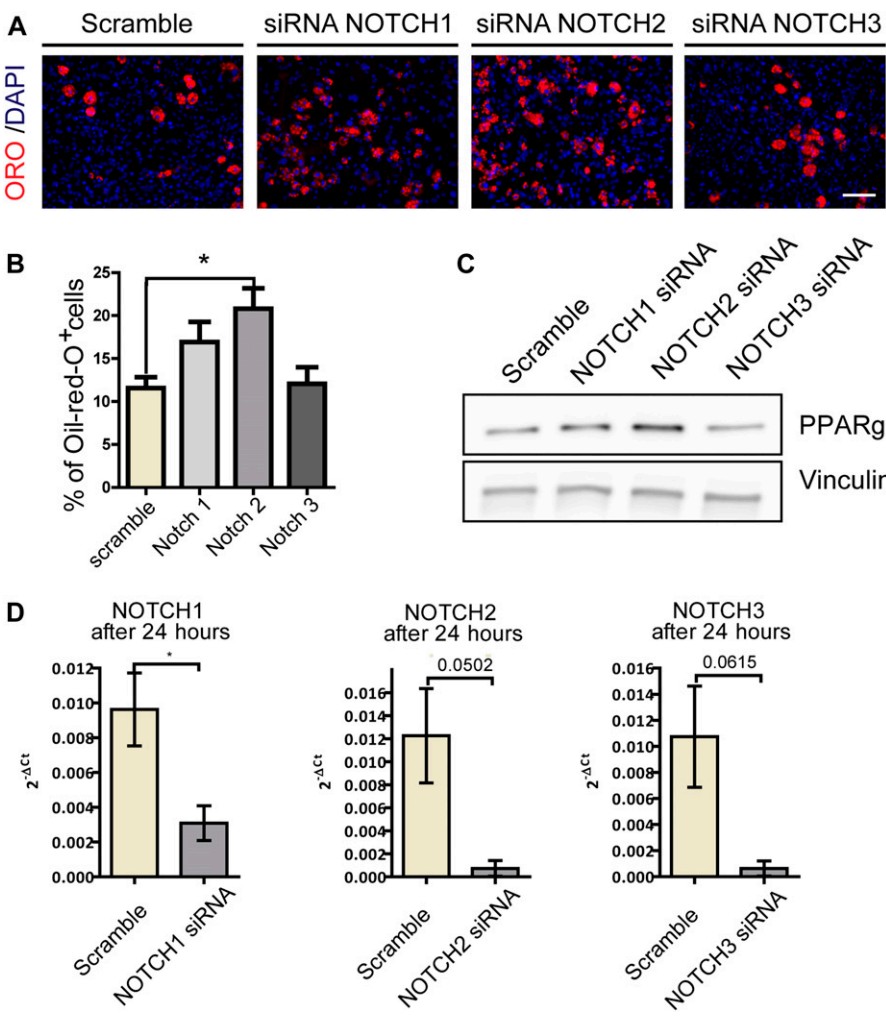

**Figure 3. siRNA knockdown of NOTCH receptors promotes FAPs adipogenesis.**
FAPs were isolated from C57BL/6 mice incubated with 50 nM of three different smart-pool of *NOTCH* (*NOCTH1, NOTCH2* and *NOTCH3*) or of a scramble siRNA used as control. To evaluate the effects of the NOTCH silencing on adipogenic differentiation, cells were maintained in adipogenic media for 3 d, 48 h after silencing. **(A)** Adipocytes were stained with ORO and nuclei with DAPI. Scale bar: 100 μm. **(B)** The percentages of ORO–positive cells are shown in the graphs (n = 3). Quantifications were done using the CellProfiler software. **(C)** Western blot analysis of PPARg expression 48 h after silencing. Vinculin was used as loading control (n = 3). **(D)** RT-PCR analysis of *NOTCH* receptors expression 24 h after silencing (n = 3).

because NOTCH signaling is transmitted via cell–cell contact, we investigated if the adipogenesis inhibition observed in coculture was mediated by activation of NOTCH signaling. To this end, we treated the SC–FAPs cocultures with 5 μM DAPT according to the protocol in Fig 4A. When FAPs were treated with DAPT in coculture conditions, adipocyte differentiation was restored, whereas satellite differentiation and myotube formation were unaffected (Fig 4E–I). In addition, we did not observe a significant difference in the number of nuclei in controls and DAPT-treated cocultures, suggesting that proliferation was also not affected (Fig 4F). These results were confirmed with Western blot analysis for myosin heavy chain (MYHC) and PPARg expression in control and DAPT-treated cocultures (Fig 4I). Taken together, these results suggest that SCs, or SC-derived myotubes, inhibit FAP differentiation by activating NOTCH.

We next asked whether NOTCH has any role in limiting adipogenesis in vivo during muscle regeneration. Cardiotoxin-injured mice were treated for 3 wk by intraperitoneal injection of DAPT starting 3 d after muscle injury. A mild fatty tissue infiltration was revealed by perilipin immunostaining only in the group treated with DAPT (Fig 4J–L).

Overall, our observations support a novel role of NOTCH in muscle regeneration, which is not only limited to modulation of SC activation and differentiation but also participates in the control of FAP differentiation. The mild pro-adipogenic effect observed after NOTCH inhibition is consistent with the existence of multiple control systems that cooperate to limit adipogenesis.

## FAPs isolated from *mdx* mice are insensitive to adipogenic inhibition by NOTCH

Building on the observation that NOTCH suppresses FAP adipogenesis in *wt* mice, we asked if the same mechanism is responsible for restraining FAPs from differentiating into adipocytes in young *mdx* mice. FAPs were isolated from 6-wk-old *mdx* mice and seeded on plates coated with the NOTCH ligand DLL1. Strikingly, DLL1-treated *mdx* FAPs did not show any significant variation in proliferation or differentiation when compared with untreated *mdx* cells (Fig 5A–C). This result suggests that, differently from *wt* mice, FAPs from *mdx* mice are insensitive to NOTCH inhibition.

Given that muscles of young *mdx* mice are characterized by repeated regeneration–degeneration cycles (Grounds et al, 2008), we entertained the hypothesis that this decreased sensitivity to NOTCH ligand could be a consequence of cell state perturbation because of chronic inflammation. To address this hypothesis, 6-wk-old *wt* mice

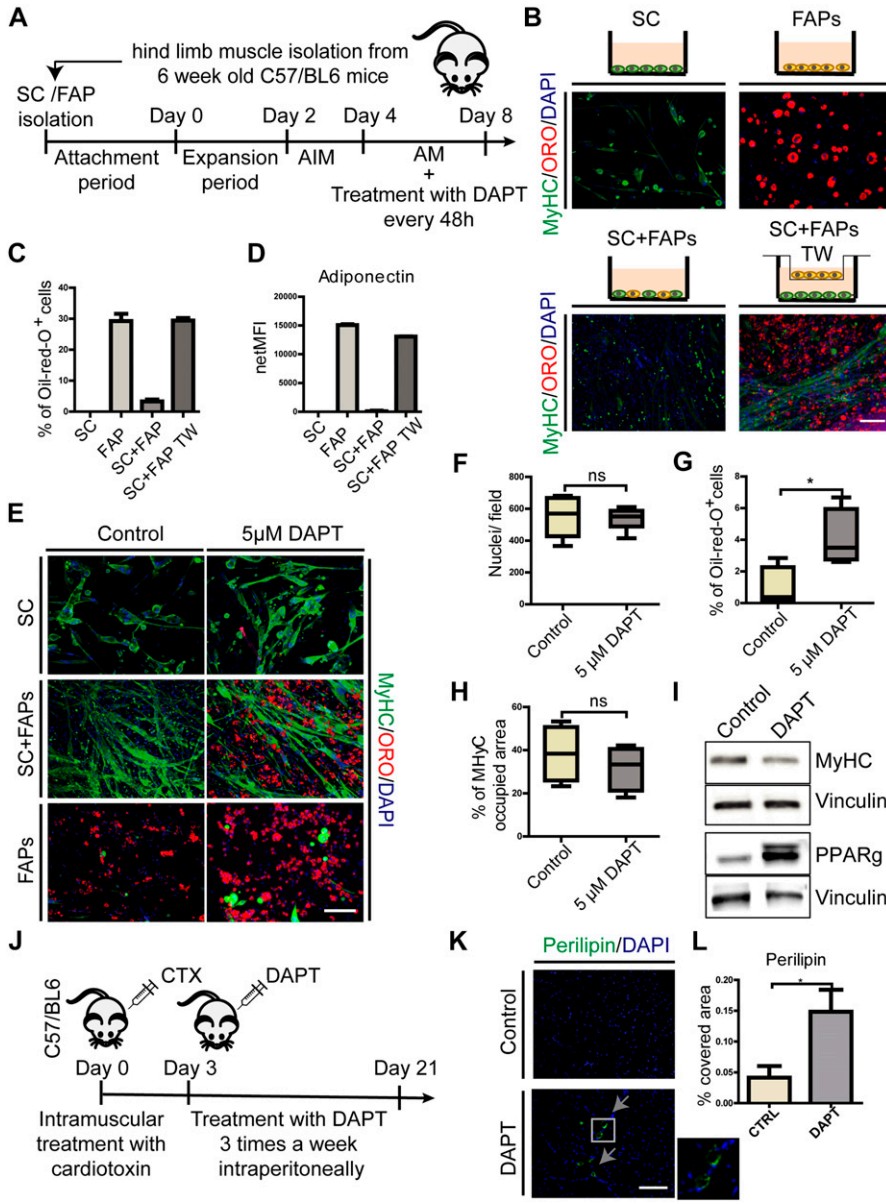

**Figure 4. Myotube inhibition of FAP adipogenesis is dependent on NOTCH activity.**
**(A)** Schema of the experimental procedure. **(B)** SCs and FAPs were cultured separately or cocultured (directly or in transwell [TW]) in SC growth medium for 2 d, followed by 2 d in AIM and 4 d in adipogenic maintenance medium (AM). Cells were stained with anti myosin heavy chain (MYHC) antibodies, ORO and DAPI. Scale bar: 100 $\mu$m. **(C, D)** The percentage of cells stained with ORO and (D) the amount of adiponectin secreted in the culture medium were quantitated and reported as bar graphs showing average ± SEM (n = 2). **(E)** SCs and FAPs were cocultured and treated with 5 $\mu$M DAPT every 48 h. Cells were stained with anti-MYHC antibodies, ORO and DAPI. Scale bar: 100 $\mu$m. **(F–H)** Box plots representing the average number of nuclei per field, the fraction of cells stained with ORO and the fraction of pixels stained with anti MYHC antibodies, respectively. Quantifications were done using the CellProfiler software. Box plots show median and interquartile range with whiskers extended to minimum and maximum values (n = 4). Statistical significance was evaluated by the $t$ test (*$P$ < 0.05). **(I)** Proteins (30 $\mu$g) from the coculture cell lysates were separated by SDS–PAGE. Western blot analysis of MYHC and PPARg expression in control and treated cocultures. Vinculin was used as a loading control (n = 3). **(J)** Schema of in vivo experiment. 6-wk-old *wt*, C57BL/6, mice were injected with cardiotoxin (10 $\mu$M) and, starting from day 3 after injury, were treated with 30 mg/kg DAPT or vehicle, intraperitoneally three times a week until day 21. **(K)** Representative TA sections of control and mice treated with DAPT stained with perilipin and DAPI (n control = 4 mice, n DAPT = 4 mice). Scale bar: 100 $\mu$m. **(L)** Graph showing quantification of perilipin covered area on TA sections, performed with ImageJ software. Statistical significance was evaluated by the $t$ test (*$P$ < 0.05).

were treated with cardiotoxin (*ctx*) to induce inflammation and regeneration in a *wt* muscle. 3 d postinjury (PI) FAPs were isolated, as at this point they reach their proliferation peak (Lemos et al, 2015). However, when seeded on DLL1-coated surface, FAPs prepared from these regenerating muscles failed to differentiate and were as sensitive to NOTCH inhibition as FAPs prepared from uninjured muscles (Fig 5D–F). These results highlight a differential sensitivity of *wt* and *mdx* FAPs to NOTCH-mediated modulation of differentiation and may be the basis of the observed fat infiltrates in aging dystrophic organisms.

Interestingly, the HES-1 protein, a downstream target gene of the NOTCH pathway, is more expressed in *mdx* rather than *wt* FAPs, suggesting that the canonical NOTCH pathway is functional in *mdx* FAPs (Fig 5G and H). Up-regulation of HES-1 is observed also in *ctx* FAPs, suggesting that the NOTCH signaling pathway is activated in

FAPs upon injury. We surmise that the insensitivity of *mdx* FAPs to the anti-adipogenic effect of NOTCH activation is not a consequence of a defect in some components of the canonical NOTCH signaling machinery but rather that the inhibition of adipogenesis is the result of a cross talk between the NOTCH pathway and an yet unidentified molecular mechanism that is affected in dystrophic FAPs.

## Remodeling of the *mdx* FAP proteome

To elucidate the molecular mechanisms underlying the different sensitivity of *mdx* FAPs to NOTCH modulation, we performed deep mass spectrometry (MS)-proteomic profiling of FAPs freshly isolated from hind limb muscles of *wt*, *mdx* and *ctx* mice. We applied the recently developed in-StageTip (iST) proteomics workflow

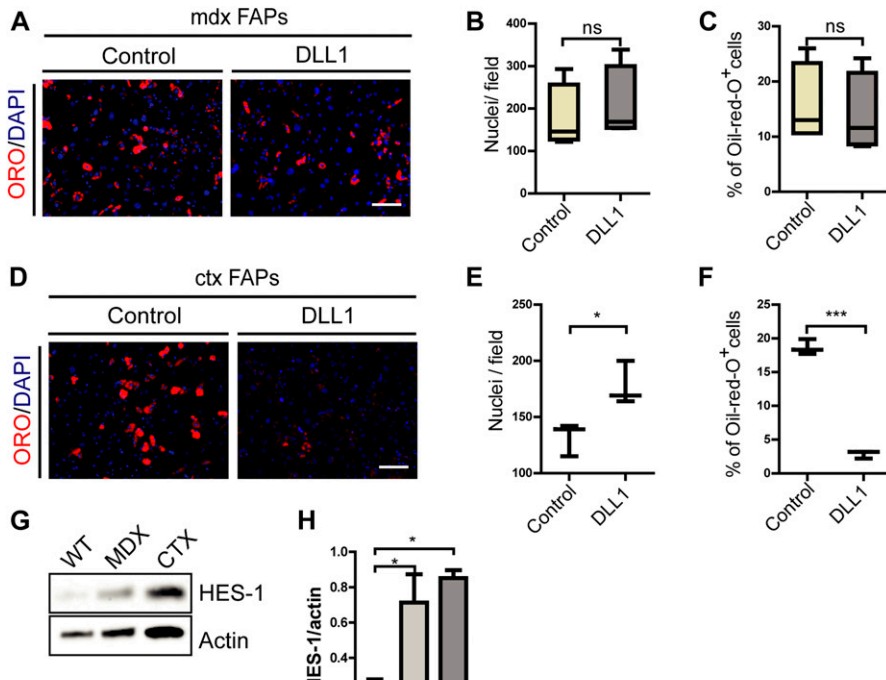

**Figure 5.** *mdx* **FAPs are unresponsive to DLL1 activation of NOTCH signaling.**
**(A)** FAPs isolated from young (6-wk-old) *mdx* mice were plated on DLL1-Fc or IgG2A-Fc (control)–coated surface in growth medium for 8 d. Adipocytes were stained with ORO and nuclei with DAPI. Scale bar: 100 μm. **(B, C)** The average number of nuclei and percentage of adipocytes are presented as box plots (n = 4). **(D)** FAPs isolated from ctx-treated C57BL/6 mice 3 d after treatment were plated on DLL1-Fc or IgG2A-Fc (control) coated surface in growth medium for 8 d. Adipocytes were stained with ORO and nuclei with DAPI. Scale bar: 100 μm. **(E, F)** The average number of nuclei and percentage of adipocytes are presented as a box plot (n = 3). All quantifications were done using the CellProfiler software. Box plots show median and interquartile range with whiskers extended to minimum and maximum values. Statistical significance was evaluated by *t* test (*P < 0.05). **(G, H)** Expression of HES-1 in FAPs from the three experimental conditions (*wt*, *mdx*, and *ctx*) was evaluated. Proteins were isolated from freshly isolated FAPs from *wt*, *mdx* and ctx-treated mice, and the expression of HES-1 was detected by immunoblotting and (H) quantified by densitometric analysis (n = 3). Actin was used as a loading control. Statistical significance was evaluated by the ANOVA test (*P < 0.05, ***P < 0.001).

(Kulak et al, 2014), combined with label-free LC–MS/MS analysis (Fig 6A), and processed the results in the MaxQuant environment (Cox & Mann, 2008; Cox et al, 2011). All experiments were performed in at least biological triplicates, revealing high quantitation accuracy and reproducibility, with Pearson's correlation coefficients ranging between 0.85 and 0.95 (Fig S6A). This approach enabled the quantification of about 7,000 proteins (Table S2), with abundance spanning a range covering seven orders of magnitude. As previously described in cell lines, structural proteins and proteins in basic cellular machineries are much more abundant than regulatory proteins, such as transcription factors (Fig S6B). To investigate whether our large-scale proteomic data could enable unsupervised classification of *mdx*, *ctx* and *wt* FAPs, we performed principal component analysis of ~4,000 proteins quantified in at least 50% of our experimental conditions. Remarkably, principal component analysis segregated the three biological conditions into three main clusters (Fig 6B), confirming the robustness of our approach.

To investigate the possible functional consequence of proteome remodeling in *mdx* and *ctx* FAPs, we focused on the 2,041 proteins significantly modulated across the three different experimental conditions (*t*-test, FDR < 0.05) (Fig 6C). This approach enabled the identification of three groups of significantly modulated proteins: (i) *mdx*-specific proteins; (ii) *ctx*-specific proteins; and (iii) proteins similarly modulated in *mdx* and *ctx* FAPs when compared with *wt* FAPs (Fig 6D–F). Next, we investigated whether these groups of proteins were enriched for specific biological processes or pathways. Interestingly, proteins involved in cell cycle and DNA replication were up-regulated in both *mdx* and *ctx* FAPs, consistent with

their activation and proliferation after chronic and acute muscle damage (Joe et al, 2010). This conclusion is also supported by the RNA expression data suggesting that genes involved in the cell cycle are controlled at the transcriptional level (Fig S4). In addition, we observed that proteins involved in the tricarboxylic acid cycle (TCA) cycle and fatty acids metabolism were down-regulated in *mdx* and *ctx* FAPs. This suggests that, similarly to SCs, FAPs activation during the muscle regeneration process may experience a metabolic switch (Ryall et al, 2015).

Next, to identify mechanisms that would explain the reduced sensitivity of *mdx* FAPs to the anti-adipogenic signals triggered by NOTCH, we investigated whether the 600 proteins that are specifically modulated in FAPs from *mdx* mice were functionally connected to the NOTCH pathway and to the adipogenesis regulatory network. The overlay of our proteomic data onto the NOTCH and adipogenesis pathways, compiled from the literature information curated in the SIGnaling Network Open Resource (SIGNOR) database (Perfetto et al, 2016), highlighted different mechanisms that may underlie the diverse response of *mdx* FAPs to NOTCH-dependent adipogenic signals: (i) the down-regulation of components of the SWI/SNF chromatin remodeling complex, known to positively modulate the transcriptional activity of NOTCH (Yatim et al, 2012); (ii) the up-regulation of the CDK5, kinase that phosphorylates and promotes the transcriptional activity of the adipogenesis master gene, PPARg (Choi et al, 2010); and (iii) the up-regulation of proteins involved in retinol metabolism including the retinoic receptor RXRA (retinoid X receptor alpha), which forms a transcriptionally active heterodimer with PPARg, driving an adipogenic differentiation program (de Vera et al, 2017).

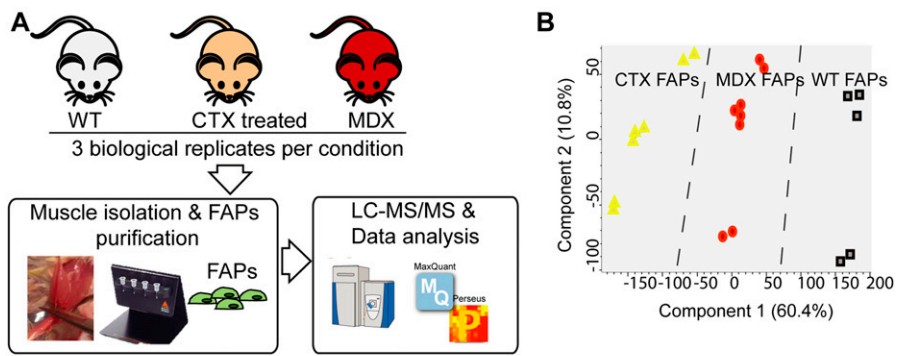

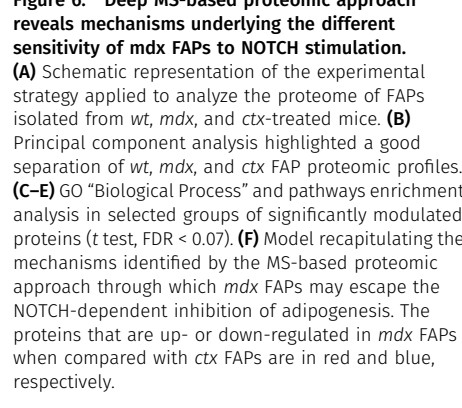

**Figure 6. Deep MS-based proteomic approach reveals mechanisms underlying the different sensitivity of mdx FAPs to NOTCH stimulation.**
**(A)** Schematic representation of the experimental strategy applied to analyze the proteome of FAPs isolated from *wt*, *mdx*, and *ctx*-treated mice. **(B)** Principal component analysis highlighted a good separation of *wt*, *mdx*, and *ctx* FAP proteomic profiles. **(C–E)** GO "Biological Process" and pathways enrichment analysis in selected groups of significantly modulated proteins (*t* test, FDR < 0.07). **(F)** Model recapitulating the mechanisms identified by the MS-based proteomic approach through which *mdx* FAPs may escape the NOTCH-dependent inhibition of adipogenesis. The proteins that are up- or down-regulated in *mdx* FAPs when compared with *ctx* FAPs are in red and blue, respectively.

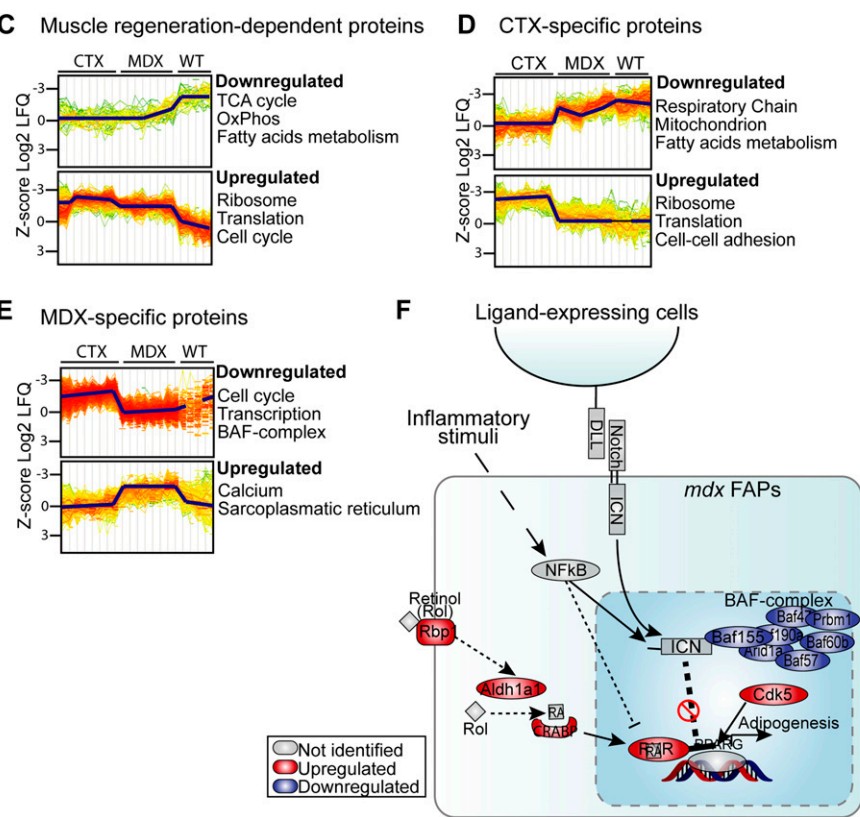

Given the importance of retinoic acids in adipogenic control (Berry et al, 2012), we focused our analysis on the *mdx*-dependent up-regulation of the RXRA receptor. Interestingly, both RXRA and NOTCH pathways are regulated by the activation of nuclear factor kappa-light-chain-enhancer of activated B cells (NFKB), a master regulator of the inflammatory response. Specifically, while NFKB inhibits the RXRA-PPARg complex through the up-regulation of the HDAC3 protein (Ye, 2008), it exerts a positive effect on the transcriptional activity of NOTCH (Ando et al, 2003).

Altogether, these observations are consistent with a model whereby the modulation of NFKB restores sensitivity to the NOTCH-dependent adipogenic signals in *mdx* FAPs. Given that inflammation activates NFKB and FAPs from *mdx* mice are chronically exposed to inflammatory cytokines, we asked whether an inflammatory environment had any effect on *mdx* FAPs differentiation.

**Synergic cooperation of muscle inflammatory response and NOTCH activation in the regulation of *mdx* FAP differentiation**

Prompted by the observation that the *mdx* proteome is perturbed in key mediators of the cross talk between the inflammatory response, NOTCH and adipogenic pathways, we asked whether an inflammatory environment had any effect on FAP differentiation. To address this, we isolated FAPs from young *mdx* mice and treated them with conditioned media (CM) obtained from CD45[+] cells from muscles of young *ctx* and *mdx* mice (Fig 7A). We could not observe any significant difference in adipogenic differentiation in cells treated with CM compared with untreated controls. However, when *mdx* FAPs treated with CM were seeded on the NOTCH ligand DLL1, we could observe significant adipogenic inhibition. This suggests a cooperation of some factor(s) produced by inflammatory cells and the NOTCH ligand

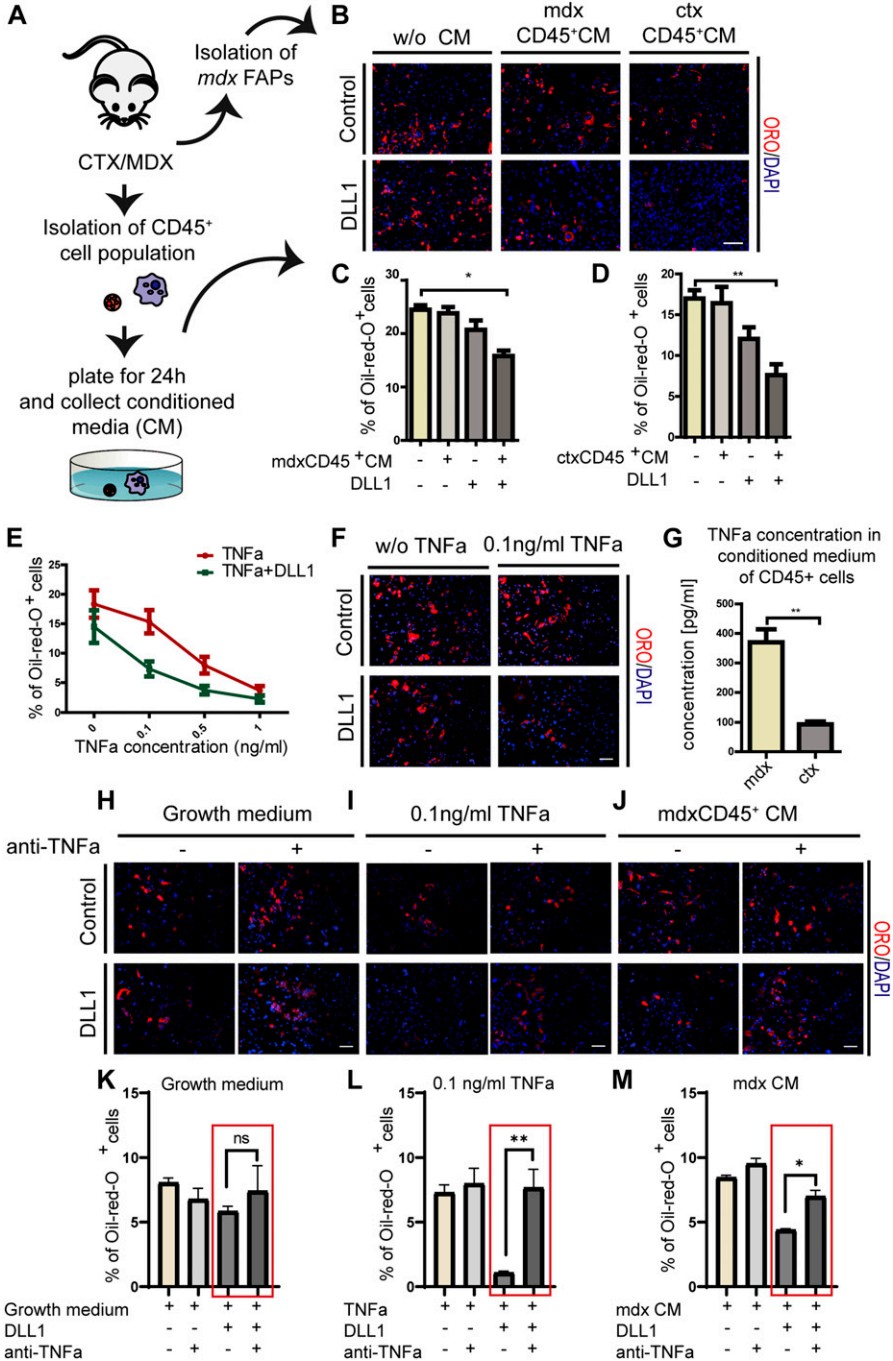

**Figure 7. Signals secreted from inflammatory cells relieve the *mdx* FAP insensitivity to inhibition of adipogenesis mediated by NOTCH activation.**
**(A)** Schema of the experimental procedure. **(B)** FAPs were isolated from *mdx* mice, seeded on plates containing DLL1-Fc or IgG2A-Fc, and treated every 48 h with conditioned media (CM) obtained from CD45⁺ cells from *ctx* and *mdx* mice. To obtain CD45⁺ CM, the CD45⁺ cell fraction was seeded for 24 h in RPMI medium containing 10% FBS. After 24 h, the CM was collected, filtered to remove unattached cells and debris, and stored at 4°C. Adipocytes were stained with ORO and nuclei with DAPI. Scale bar: 100 μm. **(C, D)** Bar plots are presenting the percentages of adipocytes of at least three different experiments. **(E)** *mdx* FAPs ± DLL1 were treated with 0.1, 0.5, and 1 ng/ml TNFa and the percentage of adipocytes was quantified. Data are represented as average ± SEM (n = 3). **(F)** Representative immunofluorescence images of adipogenesis of FAPs treated with 0.1 ng/ml TNFa in the presence of the NOTCH ligand DLL1. Adipocytes were stained with ORO and nuclei with DAPI. Scale bar: 100 μm. **(G)** Comparison of concentrations of TNFa in CM obtained from *mdx* and *ctx* CD45⁺ cells measured by ELISA. **(H–J)** *mdx* FAPs ± DLL1 were treated with anti-TNFa antibody in three conditions: growth medium, 0.1 ng/ml TNFa, and *mdx* CD45⁺ CM. Adipocytes were stained with ORO and nuclei with DAPI. Scale bar: 100 μm. **(K–M)** Bar plots representing the percentage of adipocytes. Data are represented as average ± SEM (n = 2). All quantifications were done using the CellProfiler software. Statistical significance was evaluated by the ANOVA test (*$P < 0.05$, **$P < 0.01$).

in the negative regulation of *mdx* FAPs differentiation ex vivo (Fig 7B–D). TNFa is a pro-inflammatory cytokine expressed by macrophages type 1 during the early inflammatory phase of the regeneration process (Tidball & Villalta, 2010). Its expression reaches a peak around day 3 PI (Lemos et al, 2015). We investigated whether this cytokine could be held responsible for the observed effect. FAPs from young *mdx* mice were plated on DLL1 covered plastic and treated with different concentrations of TNFa every 48 h for 8 d. In line with the hypothesis that TNFa can synergize with NOTCH in the inhibition of adipogenesis, we observed a dose-dependent inhibition of

adipogenic differentiation of *mdx* FAPs treated with different concentrations of TNFa every 48 h for 8 d (Fig 7E and F). After treatment with TNFa, the transcription factor NFKB translocates into the nuclei (Fig S7), but no apoptotic effect mediated by TNFa was detected up to concentrations as high as 100 ng/ml (Fig S8). Interestingly, even low concentrations of TNFa (0.1 ng/ml), which per se have little effect on *mdx* FAP adipogenesis, when combined with activation of the NOTCH pathway restore sensitivity to the anti-adipogenic effect (Fig 7E and F). These results parallel those obtained with CD45⁺ CM, suggesting that TNFa and NOTCH cooperate synergistically to inhibit *mdx* FAPs

differentiation. Consistently, the concentration of TNFa in the *mdx* CD45 CM, as measured by ELISA (0.36 ng/ml +/− 0.044), was comparable with that of purified TNFa showing a sizeable synergistic effect (0.1–0.2 ng/ml) (Fig 7G). Finally, FAPs were incubated with CM in the presence of the NOTCH ligand and an antibody against TNFa. Consistent with the hypothesis that TNFa is responsible for the observed cross talk between inflammatory signals and NOTCH, the anti-TNFa antibody titrates the anti-adipogenic effect (Fig 7H–M). Finally, we asked whether the NOTCH– TNFa cross talk, as observed ex vivo, was also modulating adipogenesis in vivo (Fig 8). To this end, we made use of clodronate-loaded liposomes. Systemic treatment with this preparation selectively depletes macrophages (Van Rooijen & Sanders, 1994) and limits inflammation by reducing the mRNA expression levels of inflammatory cytokines, TNFa included (Kawanishi et al, 2016). To address the involvement of inflammation in controlling adipocyte infiltrations in the muscle, we treated the tibialis anterior (TA) of *wt* mice with cardiotoxin and clodronate-loaded liposome or control liposome. The liposome treatment was repeated every 3 d. 21 d after cardiotoxin treatment, muscle sections were stained with anti-perilipin antibodies and the fraction of stained area was evaluated. In a parallel experiment, we combined clodronate with DAPT. Treatment with clodronate caused a marked increase of perilipin stain, proving the central role of macrophages in controlling adipogenesis (Fig 8). As already shown, blocking NOTCH processing by treatment with DAPT also causes fat deposition in vivo albeit to a much lesser extent. Remarkably, as observed ex vivo, the combined treatment had a more than additive effect on perilipin stain consistent with a synergic cross talk between the two control mechanisms.

## Discussion

FAPs are characterized as mesenchymal progenitor cells expressing the PDGFRa and SCA1 antigens (Joe et al, 2010; Uezumi et al, 2010). Our single-cell mass cytometry analysis shows that these surface markers identify a heterogeneous cell population, whose phenotypic profile is modulated by the muscle regeneration state. For instance, we observed two populations with significant quantitative difference in the expression of the SCA1 and CD34 antigens (Joe et al, 2010). The population with low expression levels was prominent in *wt* FAPs, whereas the population with high levels of both antigens included the majority of FAPs from regenerating muscles after cardiotoxin treatment (*ctx*). In *mdx* FAPs, both populations are equally represented. It has been documented that different levels of CD34 expression correlate with proliferation and quiescence of hematopoietic stem cells in vitro (Dooley et al, 2004), whereas SCA1 expression has been linked to control of proliferation, self-renewal, and differentiation of myoblasts and mesenchymal stem cells (Bonyadi et al, 2003; Mitchell et al, 2005; Epting et al, 2008). Our results parallel these observations: *wt* FAPs, isolated from an unperturbed and quiescent stem cell niche, have low levels of CD34 and SCA1, in accordance with their dormant state, whereas *mdx* and *ctx* FAPs, experiencing a regenerative environment, express higher levels of these two antigens.

When isolated from the muscle microenvironment, FAPs differentiate into adipocytes even in the absence of adipogenic stimuli (Joe et al, 2010; Uezumi et al, 2010). Why this differentiation potential is not manifested in vivo in the healthy muscle of a mouse is still a debated question. These observations imply the existence of anti-adipogenic signals afforded by muscle resident cell types that, in physiological conditions, restrain the FAP adipogenic potential. Evolutionarily conserved pathways, such as Hedgehog, WNT and NOTCH, have been implicated in the regulation of adipogenesis in mesenchymal stem cells (Rosen & MacDougald, 2006). Here we have shown that the suppression of NOTCH signaling via inhibition of γ-secretase or by interfering with the expression of NOTCH stimulates FAPs differentiation in a dose-dependent manner, whereas activation by the NOTCH ligand DLL1 leads to a significant inhibition of adipogenesis. Given the heterogeneity of the CD45⁻CD31⁻α7-integrin⁻SCA1⁺ cell population, it cannot be excluded that the addition of NOTCH ligand to *wt* FAPs favors the selective enrichment of a subpopulation with low adipogenic potential. However, the model invoking an inhibition of FAP adipogenic potential via NOTCH activation is more in accord

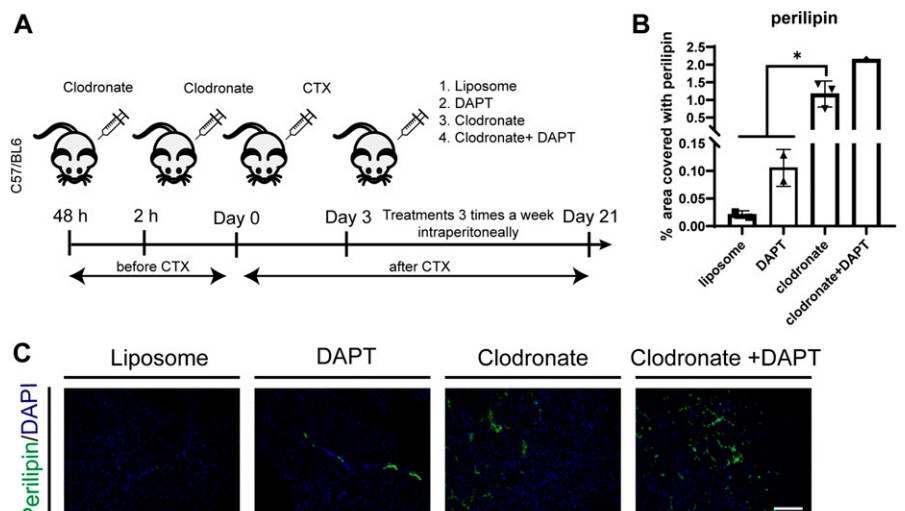

**Figure 8.   Macrophage depletion alters skeletal muscle regeneration and causes fatty tissue infiltration after muscle injury in vivo.**
**(A)** Schema of in vivo experiment. 6-wk-old *wt*, C57BL/6, mice were injected with liposomes 48 and 2 h before cardiotoxin (10 μM) injection. After 3 d from *ctx* injury, mice were treated with liposome, clodronate, DAPT, or clodronate + DAPT, intraperitoneally three times a week until day 21. **(B)** Quantification of perilipin on TA sections, performed with ImageJ software. **(C)** Representative TA sections of control (liposome) and treated mice stained with perilipin and DAPI (n liposome = 2 mice, n DAPT = 2 mice, n clodronate = 3, n clodronate + DAPT = 1). Scale bar: 100 μm. Statistical significance was evaluated by the ANOVA test (*P < 0.05, **P < 0.01).

with our data. Consistently, *ctx*-treated mice suffered mild adipocyte accumulation in the myofiber interstitial area when treated with the γ-secretase inhibitor DAPT. The observed limited adipocyte infiltrations after inhibition of NOTCH suggest that NOTCH is involved in controlling ectopic fat infiltrations in vivo while at the same time implying that additional regulatory mechanisms are likely to play a role in controlling FAP differentiation. NOTCH signaling plays a crucial role in skeletal muscle homeostasis and regeneration by regulating SC states: quiescence, activation, proliferation and differentiation (Conboy & Rando, 2002; Luo et al, 2005; Bjornson et al, 2012; Mourikis et al, 2012; Lin et al, 2013; Yin et al, 2013). Remarkably, we demonstrate that NOTCH also controls adipogenesis of muscle FAPs, adding to our understanding of skeletal muscle physiology.

A second important question relates to the reasons for the failure in the mechanisms that control FAP adipogenesis in disease conditions, as observed for instance in DMD patients or in other pathologies causing accumulation of intramuscular adipose tissue. In our work, we considered cell autonomous mechanisms that would make, with time, FAPs from a chronically inflamed muscle less sensitive to anti-adipogenic signals.

Interestingly, we demonstrated that FAPs isolated from young *mdx* mice are unresponsive to NOTCH-mediated inhibition of adipogenic differentiation when cultivated ex vivo. This finding highlights an important difference between *mdx* and *wt* FAPs, difference that could explain fat deposition in the muscles of aging *mdx* mice (M et al, 2013; Cordani et al, 2014). We hypothesized that the observed insensitivity to NOTCH might be induced by the inflammatory environment in the degenerating *mdx* muscle. Thus, we also tested FAPs isolated from a second model of muscle regeneration. However, FAPs isolated from myotoxin-treated *wt* mice are inhibited in their adipocytic differentiation when the NOTCH pathway is activated by the DLL1 ligand.

Thus, FAPs from *mdx* mice are insensitive to NOTCH inhibition ex vivo. However, the muscles of young *mdx* mice do not show adipocytic infiltrations, suggesting that adipogenesis in the muscle is under the control of multiple redundant mechanisms and that some additional form of negative control is still functional even when the one mediated by NOTCH is failing (DiMario et al, 1991; Grounds et al, 2008; McDonald et al, 2015). The change in the muscle environment caused by aging would expose the NOTCH defect and cause fat deposition.

We applied state-of-the-art MS-based proteomics to investigate the molecular mechanisms underlying the different sensitivity to NOTCH-dependent anti-adipogenic signals of *wt*, *ctx* and *mdx* FAPs. By this approach, we obtained a global picture of the proteome-wide remodeling occurring in FAPs from wild-type and regenerating muscles (young dystrophic and cardiotoxin-treated mice). Our dataset will serve as a more general resource for hypothesis-driven research to investigate the molecular mechanisms underlying functions that are perturbed in FAPs from dystrophic organisms. Overlaying MS-based proteomic profiles onto literature-derived signaling pathways has been used to extract key mechanistic information (Sacco et al, 2016a, 2016b). By this approach, we drew a network of adipogenesis-related pathways that are perturbed specifically in *mdx* FAPs and identified potential mechanisms that may explain the different sensitivity of *mdx* FAPs to anti-adipogenic NOTCH signals. Here we focused on the up-regulation of proteins

involved in retinol metabolism exclusively in *mdx* FAPs, and in particular of the RXRA receptor, which forms a complex with PPARg to positively regulate adipogenesis (de Vera et al, 2017). Our network analysis highlighted the central role of the NFKB signaling pathway that is activated in response to inflammatory stimuli such as TNFa and is known to inhibit the activity of RXRA-PPARg while enhancing NOTCH signaling (Ando et al, 2003; Ye, 2008; Maniati et al, 2011).

Considering that *mdx* mice are chronically exposed to inflammation, we further investigated the involvement of inflammatory signals in modulating *mdx* FAPs differentiation. Interestingly, *mdx* FAPs treated with CM from hematopoietic cells (CD45+ cell fraction) reacquire their sensitivity to NOTCH, and their adipogenesis is significantly reduced. TNFa is one of the main cytokines secreted by inflammatory macrophages. Because a cross talk between TNFa and NOTCH signaling has been already reported in a variety of cell types (Ando et al, 2003; Maniati et al, 2011; Jiao et al, 2012; Fazio & Ricciardiello, 2016), we hypothesized that this cytokine could assist NOTCH signaling in limiting FAP adipogenesis. In fact, we observe ex vivo that physiological concentrations of TNFa (Abdel-Salam et al, 2009) synergistically cooperate with NOTCH in the regulation of FAP adipogenesis. Overall, our results support a model whereby in young dystrophic muscles inflammatory stimuli boost NOTCH signaling and likely decrease RXRA-PPARg transcriptional activity, consequently restraining the adipogenic potential of FAPs. Macrophage recruitment to the damaged muscle has already been shown to be required for regeneration and to limit the deposition of ectopic fat (Martinez et al, 2010). Consistently, macrophage depletion by treatment with clodronate promotes adipogenic infiltrations, which are further increased by interfering with NOTCH signaling. TNFa has already been implicated as a negative regulator of adipogenesis. After TNFa antagonist treatment (etanercept and remicade), patients have an increased fat mass (Lo et al, 2007; Parmentier-Decrucq et al, 2009). In particular, Lo et al (2007) report an increase of adipose tissue in the skeletal muscles after treatment with etanercept in patients with metabolic syndrome (Lo et al, 2007). Our results add to these observations and provide evidence for TNFa-NOTCH cross talk in dystrophic muscles. It should be mentioned that several reports addressed the effect of anti-TNFa treatment in dystrophic mice and demonstrated its protective role in muscle necrosis and fibrosis (Grounds & Torrisi, 2004; Radley et al, 2008). Lemos et al (2015) suggested an antifibrotic role of TNFa in skeletal muscle, without addressing the effect of this treatment on fatty tissue infiltrations (Lemos et al, 2015). In our model, in young *mdx* mice, FAPs are insensitive to NOTCH anti-adipogenic signals, but do not cause adipocyte infiltrations in vivo, thanks to the sustained inflammatory environment (Porter et al, 2002). We have observed a substantial decrease in macrophage infiltrations in muscle of old *mdx* mice (Petrilli et al, 2017, *Preprint*), supporting the notion that, with aging, the decrease in inflammation makes the concentration of inflammatory cytokines inadequate to cooperate with NOTCH signaling, thus making adipogenesis control insufficient to prevent fat deposition.

Our results support a novel role of NOTCH signaling in skeletal muscle homeostasis. Here we propose a new molecular mechanism to explain fat infiltrations during myopathy progression. These findings may pave the way to the development of new therapeutic

strategies aimed at ameliorating the disease phenotype in muscular dystrophies.

# Materials and Methods

### Contact for reagent and resource sharing

Further information and requests for resources and reagents should be directed to and will be fulfilled by the lead contact, G Cesareni (cesareni@uniroma2.it).

### Experimental model details

#### Mice

In all the experiments, young (6-wk-old) wild-type (*wt*) C57BL/6 mice or C57BL/6ScSn-Dmd$^{mdx}$/J (*mdx*) mice purchased from the Jackson Laboratories were used. Mice were maintained according to the standard animal facility procedures, and experiments on animals were conducted according to the rules of good animal experimentation I.A.C.U.C. no. 432 of March 12, 2006.

For cardiotoxin muscle injury, *wt* mice were first anesthetized with an intramuscular injection of physiologic saline (10 ml/kg) containing ketamine (5 mg/ml) and xylazine (1 mg/ml) after which 10 $\mu$M of cardiotoxin (Latoxan L81-02) was administered intramuscularly into the TA, quadriceps, and gastrocnemius. From day 3 PI, mice were treated three times a week with intraperitoneal injection of the $\gamma$-secretase inhibitor, DAPT (95% corn oil/5% DAPT, 30 mg/kg; Sigma-Aldrich) until the 21st day PI. Control mice were treated with vehicle (95% corn oil/5% DMSO). For macrophage depletion, Clophosome-A Anionic Liposomal Clodronate and Placebo Control Liposome were purchased from FormulaMax (#F70101C-A and #F70101-A, respectively). *Wt* mice were injected with Clodronate liposomes or Placebo liposome 2 d (0.2 ml of liposomes, i.p.) and 2 h (0.1 ml of liposomes, i.p.) before cardiotoxin muscle injury. From day 3 PI, mice were treated by intraperitoneal injection with 0.1 ml of Clodronate liposomes or Placebo liposome every 3 d until the 21st day. All experimental protocols were approved by the internal Animal Research Ethical Committee according to the Italian Ministry of Health regulation.

### Method details

#### Cell isolation and MACS separation procedure

For preparation of mononuclear cell suspension, hind limbs were isolated from mice, minced mechanically and then subjected to an enzymatic dissociation for 1 h at 37°C. The enzymatic mix contained 2 $\mu$g/ml collagenase A, 2.4 U/ml dispase II and 0.01 mg/ml DNase I diluted in D-PBS with calcium and magnesium. Upon dissociation, the enzymatic reaction was stopped with Hanks' balanced salt solution with calcium and magnesium (Cat. No. 14025-092; HBSS Gibco) supplemented with 0.2% BSA and 1% penicillin–streptomycin (P/S, 10.000 U/ml). The cells suspension was subjected to sequential filtration through a 100-, 70-, and 40-$\mu$m cell strainer and centrifugations at 600$g$ for 5 min. The lysis of

red blood cells was performed with RBC Lysis Buffer (Santa Cruz Biotechnology).

SCs and FAPs were isolated from the heterogeneous cell suspension using MACS microbeads technology. For magnetic beads separation, the microbead-conjugated antibodies against CD45 (Cat. No. 130-052-301; Miltenyi Biotech), CD31 (Cat. No. 130-097-418; Miltenyi Biotech), $\alpha$7-integrin (Cat. No. 130-104-261; Miltenyi Biotech), and SCA1 (Cat. No. 130-106-641; Miltenyi Biotech) were used. SCs were selected as CD45$^-$CD31$^-\alpha$7-integrin$^+$ cells and FAPs as CD45$^-$CD31$^-\alpha$7-integrin$^-$SCA1$^+$ cells.

#### Cell culture conditions

FAPs were cultured at 37°C and 5% $CO_2$ in growth medium containing DMEM supplemented with 20% heat-inactivated FBS, 1% P/S, 1 mM sodium pyruvate, and 10 mM Hepes.

SCs were cultured at 37°C and 5% $CO_2$ on gelatin-coated plates in DMEM supplemented with 20% FBS, 10% heat-inactivated horse serum, 2.5 ng/ml bFGF (Cat. No. 450-33; PeproTech), P/S, 1 mM sodium pyruvate, and 10 mM Hepes.

For coculture conditions, cells were seeded in 1:1 ratio. For indirect cocultures, the cell culture inserts, transwells (Cat. No. 353104; Croning), with 1-$\mu$m pore were used for 24-multiwell plates. Freshly isolated FAPs were plated on porous membrane of the transwell, whereas SCs were seeded on the bottom of the plate previously coated with gelatin.

In all experiments, cell were seeded in 96-multiwell plates (3,000 cells per well) or in 24-multiwell plates (18,000 cells per well). Attachment period in all experiments was 4 d.

For adipogenic induction, two different media were used. AIM consisted of DMEM supplemented with 20% FBS, 0.5 mM 3-isobutyl-1-methylxanthine (Cat. No. I5879; Sigma-Aldrich), 0.4 mM dexamethasone (Cat. No. D4902; Sigma-Aldrich), and 1 $\mu$g/ml insulin (Cat. No. I9278; Sigma-Aldrich). Cells were maintained for 48 h in AIM, which was then replaced with the adipogenic maintenance medium (AM) consisting of DMEM supplemented with 20% FBS and 1 $\mu$g/ml insulin.

For fibrogenic induction, cells were treated with 10 ng/ml of TGFb (Cat. No. 100-21; PeproTech).

For activation or inhibition of NOTCH, WNT, and Hedgehog pathways, the following reagents were used: DAPT (Cat. No. D5942; Sigma-Aldrich), DLL1-Fc (Cat. No. 5026-DL; RnD systems), Wnt10b (Cat. No. 2110-WN-010; RnD systems), DKK1 (Cat. No. 5897-DK-010; RnD systems), Smoothened Agonist (SAG) (Cat. No. S7779; Selleck Chemicals), and Itraconazol (Cat. No. I6657; Sigma-Aldrich).

For CD45$^+$ CM, the CD45$^+$ cell fraction was seeded for 24 h in Roswell Park Memorial Institute (RPMI) 1640 medium containing 10% FBS and 1% P/S. After 24 h, the CM was collected, filtered to remove unattached cells and debris, and stored at 4°.

For neutralization of TNFa, cells were treated with 100 ng/ml LEAF Purified anti-mouse TNFa antibody (MP6-XT22; BioLegend).

#### Inhibition and activation of NOTCH signaling

For the inhibition of NOTCH signaling, the $\gamma$-secretase inhibitor, N-[N-(3,5-difluorophenacetyl)-L-alanyl]-S-phenylglycine t-butylester (DAPT), was purchased from Sigma-Aldrich and dissolved in DMSO. Depending on experimental conditions, cells were treated with different concentrations (2, 5, and 10 $\mu$M) of DAPT every 48 h for 8 d. The control samples were treated with an equal quantity of DMSO.

For NOTCH silencing, cells were plated in 12-multiwell plates (180,000 cells per well) in growth medium. After 24 h, the medium was replaced with AIM and the interference protocol was performed. The cells were incubated with 50 nM of three different smart-pool purchased from Dharmacon. Each pool is a mixture of four siRNA provided as a single reagent, able to target *NOTCH* (ON-TARGETplus Mouse *NOTCH1* siRNA #L-041110-00-0005; ON-TARGETplus Mouse *NOTCH2* siRNA #L-044202-01-0005; ON-TARGETplus Mouse *NOTCH3* siRNA# L-047867-01-0005). A scramble siRNA was used as control. Cells were harvested 24 and 48 h later for RNA and protein extraction, respectively. Moreover, to evaluate the effects of the *NOTCH* silencing on adipogenic differentiation, cells were also silenced in 96-multiwell plate (15,000 cells per well) as previously described, and after 48 h from silencing, they were maintained in AM for 3 d.

For activation of NOTCH signaling, cells were cultured in 96-multiwell plates previously coated with 50 μl/well of 10 μg/ml DLL1-Fc (Cat. No. 5026-DL; RnD systems) or control IgG2A-Fc (Cat. No. 4460-MG; RnD systems), dissolved in PBS for 2 h at RT.

### Immunofluorescence staining

Cell cultures were fixed with 2% PFA for 15 min. Cells were permeabilized in 0.1% Triton X-100 for 5 min, blocked with PBS containing 10% FBS, and 0.1% Triton X-100 for 1 h at RT, and incubated with the primary antibody for 1 h at RT. Cells were then washed three times and incubated with the corresponding secondary antibody for 30 min at RT. Nuclei were counterstained with 1 mg/ml of DAPI for 5 min at RT.

Cryosections were permeabilized with 0.3% Triton X-100 in PBS for 30 min at RT, blocked with PBS containing 0.1% Triton X-100, 10% goat serum, and 1% glycin for 2 h at RT, and incubated with the primary antibody overnight at 4°C. Samples were then washed and incubated with secondary antibody for 1 h at RT. Nuclei were counterstained with DAPI.

The following antibodies were used: mouse anti-MYHC (1:200, Cat. No. MF20; DSHB), rabbit anti-perilipin (1:100, Cat. No. 3470; Cell Signaling), rabbit anti-PPARg (1:200, Cat. No. 2443S; Cell Signaling), mouse anti-NFκB p65 (1:200, Cat. No. sc-8008; Santa Cruz Biotechnology), rabbit secondary antibody Alexa Fluor 488 conjugated (1:250, Cat. No. A-11008; Thermo Fisher Scientific), and anti-mouse secondary antibody Alexa Fluor 488 conjugated (1:250, Cat. No. A-11001; Thermo Fisher Scientific). Images were acquired automatically with a LEICA fluorescent microscope (DMI6000B).

### Oil red O and Sudan black staining

The Oil red O solution (Sigma-Aldrich) was used for detection of lipid droplets in adipocytes in cell cultures. The stock solution (0.5% filtered solution of Oil red O in isopropanol) was dissolved in ddH2O in 3:2 ratios and filtered. The cells were incubated for 5 min at RT, followed by two washings with PBS and DAPI staining.

For detection of intramuscular fat infiltrates, the Sudan Black B Lipid Stain Kit (Cat. No. VB3102; GeneCopoeia) was used according to the protocol provided by the manufacturer.

### Detection of apoptotic cells

Apoptosis of cultured *ctx* or *mdx* FAPs was detected using the In Situ Cell Death Detection Kit (TUNEL, Cat. No. 12156792910; Sigma-Aldrich)

according to the manufacturer's instructions. Quantitation of the signals was performed using the CellProfiler software. The percentage of TUNEL-positive spots was calculated as the ratio between the TUNEL-positive spots and the total number of nuclei in each field. Positive controls for TUNEL staining were pretreated for 10 min at RT with 3 μ/ml DNase I to produce enzymatic DNA fragmentation. For DNase-treated cells, the percentage of TUNEL-positive spots is the ratio between TUNEL-positive nuclei and the total number of nuclei in each field.

### Immunoblotting

Whole cell proteins were extracted with RIPA lysis buffer (150 mm NaCl, 50 mm Tris–HCl, 1% Nonidet P-40, and 0.25% sodium deoxycholate) supplemented with 1 mM pervanadate, 1 mM NaF, protease inhibitor cocktail 200X (Sigma-Aldrich), and inhibitor phosphatase cocktail I and II 100X (Sigma-Aldrich). Protein concentrations were determined by Bradford colorimetric assay (Bio-Rad). Total protein extracts (30 μg) were separated by SDS–PAGE, transferred to membranes, and incubated in blocking solution (5% milk and 0.1% Tween-20 in PBS) for 1 h at RT. Membranes were then incubated with primary antibodies overnight at 4°C. The membranes were then washed three times and incubated with anti-mouse or anti-rabbit secondary antibody conjugated with HRP (1: 2,500, Cat. No. 1721011, Cat. No. 1706515; Bio-Rad) for 1 h at RT. The blots were visualized with an enhanced chemiluminescent immunoblotting detection system. The antibodies used were as follows: anti-MYHC (1:1,000, Cat. No. MF 20; DSHB), rabbit anti-PPARg (1:1,000, Cat. No. 2443S; Cell Signaling), rabbit anti-HES-1 (1:1,000, Cat. No. 11988S; Cell Signaling), mouse anti-vinculin (1:1,000, Cat. No. ab18058; Abcam), and rabbit anti-actin (1:1,000, Cat. No. A2066; Sigma-Aldrich). Densitometric analysis was performed using ImageJ software.

### TNFa detection

Concentration of TNFa in CD45[+] cell CM was measured by ELISA technique according to the manufacturer's instructions (Mouse TNF Elisa Kit, Cat. No. 560478; BD Biosciences). The CD45[+] cell fraction from *mdx* or *ctx* mice was seeded for 24 h in 1 ml of RPMI medium containing 10% FBS and 1% P/S (10[6] per well in 12-multiwell plate). After 24 h, the CM was collected and filtered to remove unattached cells and debris. Such prepared CM was then used to measure TNFa content by ELISA technique.

### xMAP technology

The xMAP technology was used for detection of cytokines, growth factors, and chemokines in secretome of SCs, FAPs, and cocultures. Luminex assay Mouse Premixed Multi-Analyte Kit was purchased from RnD systems. The preparation of samples, standards, microparticle cocktails (antibody mix), and reagents were performed according to the manufacturer's instructions. The samples were read in technical duplicate by Luminex instrument (MAGPIX 4.2). In this work, only data obtained for adiponectin expression levels are presented.

### Single-cell mass cytometry

For mass cytometry analysis to eliminate magnetic beads, FAPs were isolated by two-step indirect labeling using PE-conjugated

antibody against SCA1 and Anti-PE MultiSort Kit, which allows removal of magnetic beads. Both antibody and kit were purchased from Miltenyi Biotec.

Freshly isolated FAPs were suspended in D-PBS without calcium and magnesium and incubated with Cell-ID cisplatin (Cat. No. 201064; Fluidigm) at a final concentration of 5 $\mu$M for 5 min at RT. The staining was quenched by adding the Maxpar Cell Staining Buffer (Cat. No. 201068; Fluidigm). Cells stained with the cisplatin were then barcoded and labeled for the detection of cell surface and intracellular antigens according to the manufacturer's instructions. Briefly, for barcoding, cells were incubated with appropriate barcodes resuspended in 800 $\mu$l of Barcode Perm Buffer and incubated for 30 min at RT. Upon incubation, cells were centrifuged and washed twice with Maxpar Cell Staining Buffer and was proceeded to staining protocol.

Cells suspended in Maxpar Cell Staining Buffer were incubated for 30 min at RT with the cell surface protein antibody cocktail (final dilution of 1:100 for each antibody). The antibody cocktail contained anti-SCA1, anti-CD34, anti-CD146, anti-CD140b, anti-CXCR4, anti-CD31, and anti-Vimentin. For intracellular antigen labeling, cells were fixed with 1× Maxpar Fix I Buffer (5×, Cat. No. 201065; Fluidigm) for 10 min at RT and then permeabilized with 4°C ultrapure methanol (Cat. No. BP1105-4; Thermo Fisher Scientific) for 15 min on ice. Cells were then incubated with intracellular protein antibody cocktail for 30 min at RT. The antibody cocktail contained anti-Caspase3, anti-pSTAT1, anti-pSTAT3, anti-pCreb, and anti-TNFa.

After cell surface and intracellular staining, cells were labeled with iridium DNA intercalator. Cells were suspended and incubated for 1 h at RT in the intercalation solution, composed of Cell-ID Intercalator-Ir (Cat. No. 201192A, 125 $\mu$M; Fluidigm) diluted 1:1,000 with Maxpar Fix and Perm Buffer (Cat. No. 201067; Fluidigm) to a final concentration of 125 nM.

Before analysis, the cell concentration was adjusted with Mili-Q water to 2.5–5 × 10^5 cells/ml. The cell suspension was then filtered with 30-$\mu$m cell strainer cap into 5-ml round bottom polystyrene tubes. Data were acquired using mass cytometry platform, of DVS Sciences (CyTOF2).

### Total proteome sample preparation and MS analyses
All samples were lysed in GdmCl buffer, boiled, and sonicated, as previously described. Proteins were digested using LysC and trypsin (1:100), overnight at 37°C, in digestion buffer (20 mM Tris–HCl, pH 8.5, and 10% acetonitrile). The obtained peptides were desalted on C18 StageTips.

### LC–MS/MS measurements
Peptides were loaded on a 50-cm reversed phase column (75 $\mu$m inner diameter, packed in-house with ReproSil-Pur C18-AQ 1.9 $\mu$m resin [Dr. Maisch GmbH]). An EASY-nLC 1000 system (Thermo Fisher Scientific) was directly coupled online with a mass spectrometer (Q Exactive Plus; Thermo Fisher Scientific) via a nano-electrospray source, and peptides were separated with a binary buffer system of buffer A (0.1% formic acid) and buffer B (80% acetonitrile plus 0.1% formic acid), at a flow rate of 250 nl/min. Peptides were eluted with a gradient of 5–30% buffer B over 240 min followed by 30–95% buffer B over 10 min, resulting in ~4 h gradients. The mass spectrometer was programmed to acquire in a data-dependent mode (Top15) using a fixed ion

injection time strategy. Full scans were acquired in the Orbitrap mass analyzer with 60,000 resolution at 200 m/z (3E6 ions were accumulated with a maximum injection time of 25 ms).

### RNA seq
For 3′-end RNAseq, FAPs were purified from *wt* and *mdx* mice (n = 4). Total RNA was extracted using Trizol according the manufacturer's recommendations. Libraries were prepared from 100 ng of total RNA using the QuantSeq 3′ mRNA-Seq Library Prep Kit FWD for Illumina (Lexogen GmbH). The library quality was assessed by using High Sensitivity DNA D1000 Screen Tape (Agilent Technologies). Libraries were sequenced on a NextSeq 500 using a high-output single-end, 75 cycles, v2 Kit (Illumina Inc.). Approximately 44 × 10^6 reads were obtained for each sample. Sequence reads were trimmed using the Trim Galore software 16 to remove adapter sequences and low-quality end bases (Q < 20). Alignment was performed with STAR 17 on the reference provided by UCSC Genome Browser 18 for *Mus musculus* (UCSC Genome Build mm10). The expression levels of genes were determined with htseq-count 19 using the Gencode/Ensembl gene model. Differential expression analysis was performed using edgeR 20. Genes with a log2 expression ratio >|0.42| (*mdx/wt* sample) difference with a FDR of <0.05 were considered as differentially expressed. Downstream analysis involving gene set enrichment and figures were performed using the Perseus software. The RNA seq data have been deposited to the ArrayExpress database (https://www.ebi.ac.uk/arrayexpress/) with the dataset identifier (E-MTAB-8040).

## Quantification and statistical analysis

### Analysis of mass cytometry data
For mass cytometry data analysis, we exploited viSNE software, a computational approach suitable for the visualization of high-dimensional single-cell data, based on t-Distributed Stochastic Neighbor embedding (t-SNE) algorithm. viSNE is used to represent single cells in a two-dimensional plot (i.e., viSNE map). In particular, each cell is represented by a point in the high-dimensional space and each cell coordinate is the expression level (intensity) of one protein measured by mass cytometer. Applying viSNE algorithm, high-dimensional space is projected into two-dimensional space preserving pairwise distances between points. The cells expressing similar levels of monitored antigens are located closely on a bi-dimensional map. In addition, a third dimension is visualized through the color, so the expression of a parameter is displayed into the viSNE map as a gradation of color of events reported (Amir el et al, 2013).

### Proteome data processing and analysis
Raw MS data were processed using MaxQuant version 1.5.3.15 (https://elifesciences.org/articles/12813; Cox & Mann, 2008; https://elifesciences.org/articles/12813; Cox et al, 2011) with an false discovery rate (FDR) < 0.01 at the level of proteins, peptides, and modifications. Searches were performed against the Mouse UniProtKb FASTA database (2015). Enzyme specificity was set to trypsin, and the search included cysteine carbamidomethylation as a fixed modification and N-acetylation of protein and oxidation of methionine as

variable modifications. Up to three missed cleavages were allowed for protease digestion, and peptides had to be fully tryptic. Quantification was performed by MaxQuant, "match between runs" was enabled, with a matching time window of 1 min. Bioinformatics analyses were performed with Perseus (www.perseus-framework.org). Significance was assessed using two-sample *t* test, for which replicates were grouped, and statistical tests performed with permutation-based FDR correction for multiple hypothesis testing. Proteomics raw data have been deposited in the ProteomeXchange Consortium (http://proteomecentral.proteomexchange.org) via the PRIDE partner repository with the data set identifier (PXD014195).

### xMAP data analysis

The secretome analysis with Luminex (Fig 4D) was performed in biological duplicate (n = 2), where n represents cells isolated from multiple mice in two distinct experiments. The samples were read in technical duplicate by Luminex instrument (MAGPIX 4.2) and analyzed with xPONENT 3.1 software.

### Image analysis

For adipogenic differentiation quantification, we exploited Cell-Profiler software (Carpenter et al, 2006), using pipelines established in our group. Average number of nuclei was determined automatically with Cell Profiler, and adipocytes differentiation was measured as the ratio of nuclei surrounded by ORO staining or positive for PPARg and the total number of nuclei per field. Quantifications were obtained by averaging the signal from 25 images. SCs differentiation into myotubes was quantified using CellProfiler software as ratio of MYHC covered area over the total area of the analyzed field.

Mass cytometry analysis (Fig 1) was performed in biological triplicates (n = 3), where n represents cells isolated from multiple mice. To obtain enough cells to perform the experiment, cells isolated from different mice were combined as one biological replicate: 15 *wt* mice divided in three biological replicates (5 mice per replicate) were used for *wt* FAPs, 6 *mdx* mice divided in three biological replicates (2 mice per replicate) were used for *mdx* FAPs, and 3 mice treated with *ctx* were used for *ctx* FAPs. In proteomics analysis (Fig 5), each biological replicate represents cells isolated from one mouse, n = 3. In in vivo experiment (Fig 3K), n represents the number of mice used (n control = 2, n DAPT = 2). In all in vitro experiments, n represents cells isolated from one mouse, in at least three independent isolation procedures. Each condition was analyzed in technical duplicate.

For statistical analysis, *t* test or ANOVA was calculated using GraphPad Prism software to determine significant differences between mean values in all experiments. Box plots show median and interquartile range with whiskers extended to minimum and maximum values. Bar graphs show mean values ± SEM. The differences were considered significant at $*P < 0.05$, $**P < 0.01$, and $***P < 0.001$.

## Supplementary Information

## Acknowledgements

This work was supported by the DEPTH project of the European Research Council (grant agreement 322749) to G Cesareni. We acknowledge Umberto Veronesi Foundation for awarding C Fuoco with post-doctoral fellowship 2019.

### Author Contributions

M Marinkovic: conceptualization, investigation, and writing—original draft.
C Fuoco: investigation and writing—review and editing.
F Sacco: conceptualization, investigation, and writing—review and editing.
A Cerquone Perpetuini: investigation.
G Giuliani: investigation.
E Micarelli: data curation.
T Pavlidou: investigation.
LL Petrilli: investigation.
A Reggio: investigation.
F Riccio: investigation.
F Spada: investigation.
S Vumbaca: investigation.
A Zuccotti: investigation.
L Castagnoli: supervision and writing—review and editing.
M Mann: funding acquisition.
C Gargioli: supervision and writing—review and editing.
G Cesareni: conceptualization, supervision, funding acquisition, and writing—original draft, review, and editing.

### Conflict of Interest Statement

The authors declare that they have no conflict of interest.

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
