## [Reviewer comments · Life Science Alliance]

Life Science Alliance

Fibro-adipogenic progenitors of dystrophic mice are insensitive to NOTCH regulation of adipogenesis

Milica Marinkovic, Claudia Fuoco, Francesca Sacco, Andrea Cerquone Perpetuini, Giulio Giuliani, Elisa Micarelli, Theodora Pavlidou, Lucia Lisa Petrilli, Alessio Reggio, Federica Riccio, Filomena Spada, Simone Vumbaca, Alessandro Zuccotti, Luisa Castagnoli, Matthias Mann, Cesare Gargioli, and Gianni Cesareni

DOI: <https://doi.org/10.26508/lisa.201900437>

Corresponding author(s): Gianni Cesareni, University of Rome Tor Vergata and Cesare Gargioli, Tor Vergata Rome University

Review Timeline:	Revision received:	2019-05-24
	Editorial decision:	2019-05-27
	Revision received:	2019-06-06
	Accepted:	2019-06-07

Scientific Editor: Andrea Leibfried

Transaction Report:

Please note that the manuscript was previously reviewed at another journal and the reports were taken into account in inviting a revision for publication at Life Science Alliance prior to submission to Life Science Alliance.

Referee 1**Summary:**

The authors have attempted to show that the NOTCH signalling pathway negatively regulates the differentiation of FAPs into adipogenic cells. This is an important topic because the replacement of contractile tissue with non-contractile intramuscular fat or collagen deposition could contribute to impaired muscle function in chronic muscle diseases. The authors expand on studies by Uezumi et al. and Huang et al. demonstrating the myogenic cell-to-FAP cell contact inhibits FAP adipogenesis, which they found is likely to occur through NOTCH signalling. Interestingly, the authors also show that FAPs-derived from mdx skeletal muscle are resistant to the NOTCH mediated anti-adipogenic control mechanism. Using advanced proteomic analysis they found that cooperation of NOTCH with inflammatory signals that activate NF-KB (like TNF) are required to restore FAPs sensitivity to NOTCH and inhibit adipogenic differentiation. The use of advanced proteomics to uncover the requirement of soluble NFKB agonists to restore NOTCH-mediated inhibition of FAPs was impressive. However, enthusiasm for the effects of NOTCH as a regulator of the differentiation of FAPs was reduced with a lack of in vivo support. The authors' conclusion that their findings suggest new therapeutic strategies to reduce fat deposition in DMD patients would also be greatly strengthened by in vivo evidence.

Major Concerns

- Figures 1 and S2 use single cell proteomic analysis to show that FAPs derived from wild type, cardiotoxin-injured and mdx muscles differentially express surface antigens, particularly CD34 and Sca1. However, it is not entirely clear how these findings relate to the rest of experimentation in this manuscript. FAPs from WT, CTX and mdx sources all differentiate into adipocytes. There was no testing whether there is a difference in the adipogenic potential of the different subpopulations (CD34HiSca1Hi and CD34LoSca1Lo) of FAPs. Therefore, it seems that this experiment is unrelated. Experiments test the effects of FAPs source (WT, CTX, mdx) on adipogenic differentiation and NOTCH mediated inhibition of FAPs adipogenesis, but without a connection to the differential expression of the surface markers from FAPs of different origins.

We understand the point made by the referee and we partly agree with him/her. There is no mechanistic connection between the observation of surface antigen expression in different FAP populations and their sensitivity to NOTCH ligand inhibition. However, we use the results of mass cytometry characterization to support a cell autonomous mechanism underlying the higher

propensity of mdx FAPs to generate adipocytes. In principle mdx FAPs could escape differentiation control either because of a defect in the environmental signals limiting adipogenesis; or because of a defect in the molecular machinery that senses these signals. In our first figure, we show that FAPs from mdx mice are in a different cell state when compared to wild type and, as such, may respond differently to environmental cues. In addition, the Cytof results represent a detailed characterization of the cell model that we are using in our studies, which has not been reported before. A PhD student in the group is now addressing functional heterogeneity in CD34+iSca1+ and has, for instance, shown that Sca1H are more adipogenic while SCA1L are more fibrogenic ex vivo. However, we feel that this characterization, albeit thought provoking, deviates from the main message of our report while opening new interesting avenues. If the referee thinks that this figure, as it is, is confusing without offering support to our conclusions we could drop it.

- Lack of in vivo support. The majority of the experimentation in this manuscript is in vitro. Although, the in vitro experiments appear to be thorough and well-designed, it is not clear if the in vitro findings translate to the in vivo system. For example, FAPS are cultured for extended periods, an attachment period followed by up to 8-days of stimulation, but it's not clear if these conditions are physiologically relevant without in vivo corroboration.

Although we understand the point made by the referee, the adipogenic differentiation assay requires at least a week. The FAP differentiation protocol, as described in our manuscript (or similar), has been used in several reports (Dong, Silva et al., 2014, Mozzetta, Consalvi et al., 2013). What we show is that the FAPs from the two mouse models, wt and mdx, behave differently in in vitro assays (Figure 5 in the manuscript). We find it unlikely that this result is an artifactual consequence of the assay time. In addition, as suggested, in the revised manuscript we have added a new in vivo experiment supporting the cross talk between NOTCH and inflammatory signals in the control of adipogenesis.

- The greatest in vivo support for NOTCH inhibiting the adipogenic differentiation of FAPs is in Fig. 3K, which shows photomicrographs of muscle sections from CTX injured muscle also treated with NOTCH inhibitor DAPT i.p. which increased perilipin and sudan black labeling. Systemic administration of DAPT will also inhibit NOTCH signaling in other cells comprising muscle, including muscle stem cells. Conboy IM et al. 2003 previously demonstrated that the administration of a Notch inhibitor impairs muscle regeneration and promotes tissue degeneration. Therefore, there is little support of an in vivo effect of Notch signaling being important regulator of FAP differentiation.

We agree that i.p. administration of DAPT will affect the whole stem cell niche. In principle other cell types could differentiate into adipocytes and cause fat deposition. However, FAPs are the major source of fat infiltrations in the skeletal muscle tissue (Uezumi, Ito et al., 2011). In our experiment, we also checked the effect of the treatment on myofiber regeneration (Fig. 1). The myofiber cross-sectional area analysis shows a slight increase of the cross sectional area in the DAPT treated group compared to control, but also a higher number of centrally nucleated myofibers, indicating that the regeneration process is still active after 21 days. These two altered readouts are most likely the consequence of DAPT treatment on satellite activation and

proliferation. These observations seem to be in contrast, at least in part, with the results of Conboy et al.

Figure 1. Additional characterization of the experiment in Figure 3k. A) Laminin and perilipin staining of muscle sections after ip DAPT injection. B) Distribution of cross sectional area. C) Comparison of the percentage of center nucleated fibers in DAPT treated and control mice.

However, it must be emphasized that the experimental conditions are considerably different. Conboy et al 2003, by measuring the fraction of center nucleated fibers, conclude that Notch inhibition in vivo negatively affects muscle regeneration. We designed our experiments to minimize the effect of Notch inhibition on myofiber regeneration. Conboy et al injected the NOTCH inhibitor 48h after injury and analyzed regeneration at an early time, 3 days later. As an additional difference, they did not damage the muscle by ctx injury but rather by piercing the muscle multiple times with a needle. Finally, they did not look at other markers of regeneration/degeneration (eg, CSA, fat infiltration, fibrotic infiltrations).

Since activation and differentiation of satellite cells requires Notch activity in several phases of the regeneration process, in order to minimize interference with the activation and proliferation phase, we administrated DAPT three days after cardiotoxin injury, and we analyzed our readouts 21 days post-injury, when the muscle architecture is restored. Because of these experimental differences, it is therefore difficult to compare the results of the two experiments. Finally, our observation of a slightly higher CSA is consistent with reports outlining that Notch inhibition causes hypertrophy (Kitzmann, Bonnieu et al., 2006, Mu, Agarwal et al., 2016).

Concerning in vivo experiment, (DAPT ip injection) we repeated DAPT injection increasing the total number of animals to n=4 for each group (control and DAPT) and quantifying the perilipin positive fat infiltration area.. We have now included these results in panel k of Fig.4.

- An experiment that included the conditional overexpression of notch intracellular domain (NICD) in FAPs using a PDGFRa-Cre system to show reduced fatty tissue infiltration in the muscle would greatly enhance the impact of this experimentation.

As a general consideration, all the experiments suggested by the referees required the sacrifice of animals, either for in vivo procedures or for purification of primary cells. Our protocols for

this project, as approved by the University ethical committee and by the Health Ministry, allows for the use of a limited number of wild type or mdx genetic background animals. Extending the number of wild type or mdx animals to be treated with the approved protocols is possible with a relatively simple procedure. However, in order to perform the experiment suggested by the referee it is necessary to introduce in the project new genetic backgrounds and/or new procedures, that had not been included in our original applications to the animal ethic committee. Thus, we would need a qualitative and quantitative extension, which, with the current Italian regulation, needs to be approved at the national level.

In addition, to our knowledge, PDGFRa-Cre and a lox mouse expressing NICD under the control of Cre cannot be easily purchased/obtained, hence, it would require constructing new transgenic mice, which would lead us beyond the time frame that we have set for this project. Moreover, if we understand correctly the experiment proposed by the referee, he/she suggests to “cure” the FAP mdx insensitivity to NOTCH ligands by forcing the expression of NCID. This would require the engineering of a mouse conditionally expressing NCID in an mdx background with considerable extra breeding/crossing/testing. Finally, our in vivo readout requires growing the animals for at least a year when fat infiltrations can be observed in mdx mice. The FAPs that we use in our ex vivo experiments are purified from 6 weeks old mice when FAPs are already insensitive to NOTCH inhibition but no fat is observed because of the compensating effect of the inflammatory environment (Figure 7 in manuscript).

Minor Concerns:

- How long is the 'attachment period' between the FAPs isolation and cell stimulation'? Is this normalized between all groups? Do FAPs from WT, CTX or mdx attach differently?

We apologize for not providing this information. The attachment period in all experiments was 4 days. As far as we can judge, FAPs from wild type and mdx mice attach equally to plastic although this is difficult to evaluate with precision because the FAPs from mice of the two different genetic backgrounds have a different proliferation rate in the “attachment period”. However, we quantified and compared the number of wt and mdx FAPs at day 0, which is the 4th day after seeding, and at day 8. We did not observe significant difference in cell number. In addition, we measured the percentage of proliferating cells with EdU proliferation assay at day 0 and did not observe significant difference between wt and mdx FAPs (Fig.2).

Figure 2. Comparison of wild- type and *mdx* FAPs number in *in vitro* assay at day 0 (after 4 days from plating) and day 8; and the percentage of proliferating cells at day 0.

- Results, subsection 2, 1st paragraph, last sentence. "Cell proliferation" and "survival" were not assayed. Nuclei/field was assayed, which would reflect differences in cell accumulation. Although no change in cell accumulation likely means no change in cell proliferation or survival, the conclusion is not proven definitively.

We thank the referee for pointing this out. We now rephrased the text “...without affecting cell number”.

- Results, subsection 2, 2nd paragraph, last sentence. 'We conclude that FAPs, by synthesizing both a Notch ligand and receptor are able to limit their own differentiation by an autocrine mechanism.' There is limited experimental support of this. These conclusions are based on a difference in adipogenic differentiation in the presence of Notch activators and inhibitors. The authors did not look at Notch receptor or ligand expression in FAPs.

We thank the referee for his/her suggestion. We now evaluated the expression of the members of Notch signaling pathway by western blotting (Fig. 3). Since the expression level of these membrane proteins is relatively low, we could not detect them in the proteomic experiments. However, we have now performed a new RNAseq experiment to characterize the transcriptome of wt and mdx FAPs. These data are now included as supplementary materials. As shown below, our data show that FAPs express both NOCTH ligands and receptors and have therefore the potential to autoregulate their differentiation via an autocrine mechanism.

Figure 3. Notch signaling pathway in wt and mdx FAPs. A) Transcriptomics analysis of Notch receptors and Jag1 ligand expression in wt and mdx FAPs. B) Western blot analysis of Dll1, NCID and HES1 expression in wt and mdx FAPs.

- Supplemental Figure 3, I would be interested to know if NOTCH, WNT and HEDGEHOG activators and inhibitors affected the fibrogenic cellular fate of FAPs as well.

The main question of this study was the effect of Notch signaling on adipogenic differentiation of FAPs. Nonetheless, we did analyze fibrogenic differentiation of FAPs upon activation/inhibition of the Notch signaling pathway (Fig. 4) and we observed an increase in aortic smooth muscle actin (SMA) expression in the group treated with Notch ligand. SMA is a common marker of myofibroblasts, activated form of fibroblast responsible for transient production of extracellular matrix during tissue regeneration after acute injury (Mann, Perdiguero et al., 2011).

Figure 4. Expression of SMA in FAPs after activation/inhibition of the Notch signalling pathway.

In order to evaluate the effect of the WNT and Hedgehog pathway perturbations on fibrogenesis we would require the sacrifice of more animals. Since WNT and HH modulation of FAP differentiation was not the central part of the current work we decided not ask the animal committee approval for extension of the protocol.

- Results, subsection 4, 2nd paragraph. Inconsistent reference style. Grounds et al. and Lemos et al. are not numerically annotated.

We thank the referee for pointing this out. We corrected it.

- The authors do a nice job of showing that TNF when synergistically combined with NOTCH signaling inhibits mdx FAPs differentiation. Also, I agree that the source of TNF is likely to be inflammatory cells. Yet, I would like to know if FAPs from WT, CTX or mdx differentially express TNF and could regulate adipogenic differentiation in an autocrine or paracrine manner.

To respond to the referee's comment, we analyzed TNF α expression in FAPs from wt, mdx and ctx treated muscles by single cell mass cytometry, and identified a small subpopulation which is positive for TNF α and pSTAT1 (Fig. 5 and 6). Interestingly, this population was more abundant in the FAPs purified from animals treated with cardiotoxin three days post injury. Three days post injury corresponds to the peak of TNF α expression during muscle regeneration (Lemos et al. 2015). However, given the small size of the FAP subpopulation expressing TNF α , it is unlikely that this represents a major contribution to TNF α levels in the skeletal muscle. In Fig. 7 is reported an experiment where we treated mdx FAPs seeded on DLL1 in growth medium with

anti-TNF α antibodies and we observed a slight increase in adipogenic differentiation, consistent with autocrine regulation mediated by TNF α . However, this difference was not statistically significant.

Figure 5. viSNE analysis of TNF α expression in a subpopulation (circled) of wt/mdx/ctx FAPs. The expression intensity of the different antigens in each cell is mapped to a blue (low) to red (high) color scale.

Figure 6. Quantification of TNF α expression in three biological repeats of FAP preparations from wt, mdx and ctx mice.

Referee 2

The authors demonstrate that Notch signaling inhibits adipogenic differentiation of FAPs, and furthermore provide evidence that the cells that normally provide Notch signaling are myoblasts (cultured satellite cells) and fibers (myotubes). They then further show that FAPs isolated from mdx muscle, but not FAPs from cardiotoxin-injured muscle are insensitive to Notch-mediated inhibition of differentiation - they differentiated into fat even in the presence of DLL1. The authors then perform a proteomic analysis of FAPs from the 3 conditions and hone in on inflammatory signaling from TNF-alpha to suggest that reduced TNF-alpha signaling can restore the sensitivity of mdx FAPs to Notch signaling, and suppress their adipogenesis.

We are not sure whether here there is a misunderstanding or, more simply, it is a misprint. Actually, we show that, in order to restore the sensitivity of mdx FAPs to NOTCH signaling we need to “ACTIVATE” TNF signaling (Fig 7e).

These results are intriguing and will be of wide interest. However, the current data does not yet rigorously support this interpretation. The data at the end of the paper on TNF-alpha make the paper weaker, and since the inhibitory effect of Notch signaling on adipogenesis has been shown by several publications, the most significant aspect of this work is the fact that *mdx* FAPs are insensitive to Notch signaling. The authors don't actually provide a mechanism to explain this, but I think the basic result is interesting enough without a full explanation for why. However, it does then need to be very well supported.

Key criticisms:

1. For the characterization of FAPs in the muscle-injured state, an alternative explanation is that another cell population acquires *Sca1* positivity after injury or in the *mdx* mouse. Therefore, the FAP identity of the purified cells needs to be compared under the 3 conditions. It is of particular interest to evaluate PDGFRalpha levels, as this is the best marker for FAPs. It was the original marker used in the Uezumi paper, and the Joe paper also demonstrates that PDGFRa is expressed on FAPs. Since that time, studies have used *Pdgfra-cre* to mark FAPs. (1) What is the status of PDGFRa vis a vis the negative markers and *Sca1* in WT vs injured/*mdx*? If the authors assumptions are correct, PDGFRa needs to strongly correlate with *Sca1*. Please evaluate.

We agree that SCA1 is expressed in different cell types, although, PDGFRa is also expressed on additional skeletal muscle cell populations: i.e. PICs (Pannerec, Formicola et al., 2013) and a subpopulation of pericytes (Birbrair, Zhang et al., 2013). In addition, both Sca1 and PDGFRa are equally accepted as markers of FAPs and widely used for their isolation. There is a wide range of publications using FAPs isolated from both wt or mdx mice according to SCA1 expression (Heredia, Mukundan et al., 2013, Madaro, Passafaro et al., 2018) (Kopinke, Roberson et al., 2017). During the isolation process, FAPs are routinely selected from CD45/CD31/a7integrin negative populations, which minimizes the possibility to include other cell population positive for SCA1. We further confirm the identity of our cell preparation in Figure S1b showing that >86% of our cell preparation from wt FAPs is PDGFRa positive. To further support the conclusion that our cell preparation is positive for both markers regardless of the source (wt or mdx) we performed FACS analysis on FAPs isolated from mdx mice and demonstrated that >90% of cells are both SCA1 and PDGFRa positive (Fig. 7).

Figure 7. FACS analysis of Sca1 and PDGFR α expression in mdx FAPs.

2. A major part of the interpretation hinges on the activity of DAPT, the gamma secretase inhibitor that blocks Notch signaling. All small molecules have off-target effects, therefore at least one other, chemically unrelated gamma secretase inhibitor should be tested, and the similar phenotypes confirmed.
3. Gamma secretase targets a diverse set of intramembrane proteins, not just Notch. While gamma secretase inhibition clearly blocks Notch, it does a lot of other things. To rigorously demonstrate that Notch signaling is necessary, a genetic loss of function experiment is necessary. Is in vitro adipogenesis improved when Notch is knocked down or knocked out, as it is when cells are treated with DAPT?

We thank the referee for the comments. We will address points 2 and 3 together:

We agree that g-secretase targets diverse membrane proteins. However, please consider that our conclusion that NOTCH is involved in the process is based on both inhibition (DAPT) and activation (DLL1) of the pathway. To further support the conclusion that the Notch pathway controls FAPs adipogenesis, we performed in vitro knockdown of the Notch receptors, using smart pool siRNA targeting Notch 1, 2 and 3 (Fig. 8). While knockdown of Notch3 did not have any effect on adipogenic differentiation and knockdown of Notch1 slightly increased adipogenesis, the downregulation of Notch2 led to a significant increase of FAP differentiation. This approach is now discussed in the manuscript. Overall, our results support the conclusion that the Notch signaling pathway negatively controls FAPs adipogenic differentiation.

Figure 8. siRNA mediated knockdown of Notch2 receptor significantly downregulates FAPs adipogenesis.

- The experiment supporting the role of Notch signaling in repressing adipogenesis in vivo was fraught. Injecting DAPT into mice will block Notch signaling everywhere, with possible consequences in hematopoietic, myogenic, endothelial or other cells of muscle. These may have follow on effects on FAPs, causing them to differentiate into fat at a greater rate in vivo. In addition, the effect was extremely modest, and not quantified.

Although we agree that NOTCH signaling may have an indirect effect on FAPs, our in vitro experiments support the model of a direct effect of NOTCH on FAPs adipogenesis. Additionally, our new transcriptome and western blot data clearly show that FAPs express both the NOCTH ligands and receptors, further supporting that DAPT treatment affects the autocrine NOTCH signaling in FAPs. Moreover, we have now repeated the experiment to monitor adipocyte infiltrations in skeletal muscle tissue upon DAPT treatment. Specifically, we have now increased the total number of animals to n=4 for each group (control and DAPT) and quantified the area showing fat infiltration marked with perilipin. This analysis is now presented in panel L of Fig. 4. A new sentence was added in the discussion section to clarify this point.

It will be essential to use a genetic approach to selectively ablate DLL1 in myofibers and to quantify fat infiltration. If the authors do this and find that the consequence is fatty accumulation in skeletal muscles, the mechanism, and importantly its in vivo relevance, will be supported.

Concerning the ablation of Notch signal, we are not sure which of the many NOTCH ligands is (are) responsible for limiting adipogenesis in vivo. Ablating DLL1 in myofibers may be not sufficient to abolish NOTCH signaling. In addition, as correctly stated by the referee him/herself, NOTCH activation plays a major role in proliferation and differentiation of satellite cells. By interfering with it, one is likely to cause a pleiotropic effect on muscle regeneration that may be difficult to interpret. Furthermore, although we have shown that direct contact between myofibers and FAPs inhibits differentiation we have not proven that myofibers (only) are the source of “the ligand” in vivo. We need to consider that FAPs reside in the interstitial space and are separated from myofibers by the basal lamina. Additionally, as we have shown, FAPs themselves produce NOTCH ligands and autocrine mechanism could play an important role in the control of adipogenesis.

5. The authors show that HES1 is expressed in FAPs from mdx or CTX-injured mice as evidence that the Notch signaling pathway is active in these cells. This is not sufficient as it is essential to compare +/- ligand to make this conclusion. Since the authors have an assay based on DLL1-coated plates, they ought to culture FAPs from the 3 conditions in the presence and absence of DLL1 and assay key Notch target genes (like Hes1). Only if they see an increase in the presence of DLL1 can they say that the Notch signaling pathway is intact and functional in these cells.

We tried this approach but we were not successful, because we faced some technical issues - the amount of cell material that we could recover this time from our “standard” NOTCH ligand assay was not sufficient for a western analysis and the identification, for instance, of HES1. Of course, we could request the permission for sacrifice of additional animals and scale it up but we wonder whether the contribution of this experiment to the message of our work is worth the extra cost of purified DLL1. If the referees think that this experiment is absolutely necessary we could have a go with the scaling up. On the other hand, both transcriptomics and proteomics show that the levels of the main players in the NOTCH pathway are not altered. However, we agree that it is not sufficient to conclude that the NOTCH pathway is functional. Because of the lack of solid evidence on the “functionality” of the NOTCH pathway in mdx FAPs, we deleted the relevant sentence in the discussion.

6. The studies on TNF-alpha are the most problematic of the study:
The authors evaluate TNF-alpha secreted by surviving CD45+ cells isolated from muscle of WT and mdx mice. However, this a very indirect way of supporting the notion that mdx muscle has more TNF-alpha. The authors need to measure TNA-alpha directly in mdx muscle. There are ways of doing this directly.

It was not our intention to show that mdx muscles have more TNFa than wt. We did not measure TNFa directly in mdx muscles, as there is enough literature evidence showing that this inflammatory cytokine is expressed in myopathies (Kuru, Inukai et al., 2003). However,

stimulated by the referee's comment, we measured Tnfa mRNA levels in wild-type and mdx muscles confirming that in the mdx muscle the Tnfa gene is more expressed (Fig. 9)

Figure 9. Expression of TNFα mRNA in skeletal muscles of wild-type and mdx mice.

7. The result contradicts the model. If TNF-alpha restores sensitivity to DLL1, why do mdx FAPs, which are presumably exposed to more TNF-alpha, then make fat so well? DLL1 should inhibit them better than WT FAPs.

Probably this point has not been reported sufficiently clearly. Our model is “6 weeks old mdx mice”. At this age muscles do not have fat infiltrations. Thus, we have to explain why mdx FAPs, which ex vivo are “insensitive” to DLL1, are still under differentiation control in vivo. We propose that the NOTCH “ineffectiveness” is compensated by the inflammatory response (TNFα) (Figure 7). In fact, fat infiltrations occur at a later stage, in aged (12 months-old or older) mdx mice, when the number of TNFα producing macrophages decrease (ref. Petrilli LL et al, 2017 in our reference list) and become insufficient to compensate the NOTCH defect. The new experiments where we deplete macrophages by clodronate treatment are consistent with this interpretation.

8. The authors treat supernatants that inhibit adipogenesis with anti-TNF-alpha and claim that this blocks the activity. While anti-TNF-alpha clearly blocks the activity of added TNF-alpha (Fig. 6L), it does not block the activity of mdx conditioned medium (Fig. 6M). What the authors actually observed was that mdx conditioned medium resulted in a 20-25% reduction in oil red O staining, even in the presence of anti-TNF-alpha. This reduction was apparently not statistically significant, so the authors use incorrect statistical thinking to claim that this is evidence that anti-TNF-alpha eliminated the sensitivity to DLL1. Not the case! It is lack of evidence of an effect, which is something entirely different from evidence of a negative effect. The effect size is still considerable. Significance is highly dependent on sample size. If the authors had used a greater sample size, this effect would probably have been significant. What they actually need to compare is the difference between DLL1 treated wells with no anti-TNF-alpha and DLL1 treated wells with anti-TNF-alpha. If this difference is significant, they are free to claim

that the anti-TNF-alpha is diminishing the activity. In any case, this result still contradicts the model in my opinion.

We thank the referee for his/hers comment. Now, the graphs show the statistically significant difference between DLL1 and DLL1+anti-TNF α in the groups treated with TNF α and the conditioned medium, while there is no significant difference in the group with growth medium. We apologize for not having commented on the statistical relevance of this difference.

I think the paper would be much better if Figure 6 and the mass spec data were removed and used for another paper. As it is, the authors have three separate stories that do not stick well together, with one of the stories (TNF-alpha) really not filling in well, in addition to not being strongly supported.

We do not agree with this conclusion and we hope that our rebuttal convinced the referee that the mass spec data has allowed us to home in the inflammatory response as an important component that synergize with NOTCH in controlling adipogenesis in the muscle.

We performed an in vivo experiment to provide additional support to our conclusion that TNF α and NOTCH synergy are important in regulating FAPs adipogenesis (Fig. 10). Briefly, we first injured the muscle with cardiotoxin and depleted the animals of macrophages, a main source of TNF α , by administering clodronate-loaded liposomes (Kawanishi, Mizokami et al., 2016) every 3 days starting from the 3rd day post-injury (time point corresponding to the peak of TNF α expression). As a control, we used PBS liposomes. In order to evaluate the synergic effect of inflammatory cues and NOTCH signaling a group of mice treated with clodronate liposomes was additionally treated with DAPT, 3 times a week. The animals were sacrificed after 21 days and muscle sections were stained for perilipin. The fraction of fat perilipin positive stained area was quantified using ImageJ.

Figure 10. In vivo depletion of macrophages and Notch inhibition leads to increased fat tissue deposition in wt mice after CTX injury.

We observed a significant increase of fat infiltrations in the macrophage depleted group, confirming the importance of inflammation in muscle regeneration. However, when mice were additionally treated with DAPT, we could observe an additional increase of fat infiltrations. Unfortunately, 2 of the three mice treated with both drugs, died and prevented us from performing a statistical analysis. Nonetheless, these data clearly show the involvement of macrophages in regulation of adipogenesis in mdx muscles. We hope that this analysis, combined with the results of the in vitro experiments, provides enough information to convince the referee of the TNF α -NOTCH synergy in regulation of FAPs adipogenesis.

Additional issues:

- Please please describe the CTX injury protocol in the methods and indicate on which day post-CTX injury the FAPs were analyzed in the figure legend (Fig. 1). 3 days post-injury is mentioned in the results section only.
- Please define TW (trans-well?) in Figure 3B.
- The legend to Figure 6K, L, and M is missing.

We thank the referee for pointing this out. We made the relevant changes.

Referee 3

First, this reviewer apologizes for the slow review. This time of year is difficult and several papers were under review at this time.

The study explores how FAPs, an interstitial cell population that has been shown to underlie fibrosis and fatty infiltration in skeletal muscle pathologies, are potentially altered in mdx (murine Duchenne) muscle. The premise is very exciting and the topic well worth considering for publication in this journal. The salient points demonstrated are that Faps from mdx deficient mice have different cell fate potentials and correspondingly, different cell surface signatures as determined using mass spec. They show that all FAPs, regardless of whether they are obtained from uninjured, injured or mdx muscle are able to differentiate into adipocytes (NB the authors should refer to these cells as pre-adipocytes). By using an inhibitor of gamma secretase, they demonstrate that Notch signaling is required for adipocyte differentiation (i.e. Notch inhibits adipogenesis). While not a major point, this inhibitor is not entirely specific for the Notch pathway and thus several other non-pharmacological approaches should have been used such as siRNA directed more specifically to the Notch pathway.

We have now performed the RNA interference knock down as suggested by the referee.

Concerning the pre-adipocyte to adipocyte terminology issue we must admit we are not adipogenesis experts. However, we read (Rosen & MacDougald, 2006):

“The first phase, known as determination, involves the commitment of a pluripotent stem cell to the adipocyte lineage. Determination results in the conversion of the stem cell to a pre-adipocyte, which cannot be distinguished morphologically from its precursor cell but has lost the potential to differentiate into other cell types.”

The cell that we call adipocytes can be clearly distinguished from their precursors. Thus, at the moment we have left the term “adipocyte”. However, we are open to discussion.

This stated, they do explore this issue further with Notch ligand and show that adipogenesis of FAPs and myogenesis by myoblasts can be uncoupled in co-culture experiments.

A central point and observation of this study is that they show that FAPs isolated from mdx muscles (at a time corresponding to degeneration of the muscle) are insensitive to Notch where as wildtype FAPs from injured muscles are sensitive. The others provide some evidence that this insensitivity is not due to the lack of a Notch signaling component (effector) but this issue is not completely explored. They do show that activated FAPs and mdx FAPs show a metabolic activation switch consistent with previous studies on other stem cells. Again, this is not completely explored but of interest. Nonetheless, the mdx derived FAPs are Notch insensitive and the authors suggest that this may be due to the inflammatory environment of FAPs in mdx muscle. Consistent with this notion, they show that NFkB is activated consistent with an NFkB upregulation of NOTCH signaling.

The experiment they perform to test if the inflammatory environment has an effect may not be the best way to do it. Rather, instead of taking 'young' FAPs from mdx deficient mice and exposing them to CD45+ CM from cdx or mdx mice, it would make more sense to expose normal FAPs to these conditions, at least as a control.

We have thought about this. However, we believe that the experiment, as outlined by the referee, is not informative. Wt FAPs are already sensitive to NOTCH ligands, thus we will not be able to observe a rescue of the antiadipogenic effect.

In addition, isolating FAPs from another model of chronic muscle injury (or simply repeated injury) might show that FAPs become refractory to Notch signaling. Nonetheless, the notion that TNF activates the NOTCH responsiveness in FAPs is logical and follows well with the notion that NFkB 'translocation' (not migration as the authors refer to this process) to the nucleus is induced by low levels. The authors should show that in addition to nuclear translocation, there is indeed NFkB activation and there are several assays to routinely verify this.

We thank the referee for this comment. We rephrased the text by replacing "migration" with "translocation". We explored the possibility to add an independent evidence of NFkB activation as suggested by the referee. However, we had difficulty to identify a reliable assay that was compatible with the relative low amount of cell material that we use in our experiments. On the other hand, I think we do not need to show that exposure to TNF α activates NFkB. There is plenty of support in the literature. Furthermore, translocation of p65 into the nucleus is widely used in the literature as evidence of NFkB activation (Trask, 2004).

What is centrally missing in this study that would make this reviewer more enthusiastic is a more mechanistic understanding as to why the FAPs have changed? If mdx FAPs are injected into healthy muscle, will they function appropriately? If FAPs are derived from a conditional NOTCH mutant, recombined and injected into healthy muscle, would they partially recapitulate the fibrotic and fatty phenotype?

We are not sure whether FAPs, once they are taken from their natural niche and re-injected into muscle, will still be under appropriate adipogenesis control. Ideally to support our model we should have to show that wild type FAPs re-injected into muscles (wild type or mdx) do not differentiate while mdx FAPs only differentiate when injected into old mdx mice and do not when injected into young mice (wt or mdx). In order for it to be approved by the ethic committee, we would have to justify this experiment on a solid scientific ground and account for the sacrifice of more animals that we had originally anticipated. Nevertheless, we performed some preliminary experiments by labelling purified FAPs with a lentivirus expressing b-galactosidase and injecting them into mouse muscles. The re-implant efficiency was very low and this discouraged us from attempting more complex experiments that required larger amounts of cells.

The single in vivo analysis using DAPT are not too convincing. Results should be quantified rigorously and not just presented by selected photomicrographs.

We thank the referee for the comment. We performed new experiments and updated Figure 3K with quantification of perilipin-covered area in muscle sections.

In addition to these major points, the authors raise the point in the discussion as to why FAP adipogenic differentiation does not occur in healthy muscle. The authors should keep in mind that mice are unusually lean compared to other mammals and many muscles, particularly from

larger mammals, have normal interstitial fat depositions (i.e. a 'ribeye' steak).

Thanks, we missed this point. We now specify in the sentence that we are talking about mouse muscles.

Figure 2, panel C-poor quality. Needs to be improved with high resolutions panels as well.

We have now added to the panels high resolution pictures.

Anti TNF experiments on mdx mice have been published and should be discussed in the context of this work. Does remicade treatment lead to a change in fatty infiltration?

We apologize for not citing these reports in our discussion. We have now revised the discussion and included this information. However, most of the published work investigated the effect of anti-TNF α treatment on fibrosis in mdx mice and did not look into fatty tissue infiltrations (Ermolova, Martinez et al., 2014, Grounds & Torrisi, 2004, Radley, Davies et al., 2008). However, in order to demonstrate the involvement of inflammatory cues in regulation of fat deposition in mdx muscles we performed a new in vivo experiment after macrophage depletion. Briefly, we induced an injury in muscles that were depleted of macrophages by clodronate treatment. We could observe a significant increase in fat infiltrations in treated muscles. This effect was even more pronounced when mice were additionally treated with DAPT, NOTCH inhibitor (Fig. 10). Even though we did not specifically target TNF α , this observation together with ex vivo evidence, support the role of TNF α in regulation of fatty infiltration in the skeletal muscle tissue.

In general, there is some critical missing in vivo work here. While it is a significant demand to show that blocking Notch signaling via conditional mutants in FAPs would confirm the conclusions in this study, short of this, blocking this pathway in vitro using other than pharmacological approaches would be a minimal threshold (i.e. siRNA) as well as the engrafting experiments indicated above.

We thank the referee for his/her suggestion. We performed in vitro knockdown of Notch receptors using smart pool siRNA, and demonstrated that downregulation of the Notch2 receptor leads to significant increase in FAPs adipogenic differentiation. Please see response to referee 2 for more details. In addition, we supported the in vivo relevance of the NOTCH- inflammation cross talk by the macrophage depletion experiment.

This stated, the premise of this study is very exciting, and while the mass spec is understandably an important resource for the research community, the study, in toto, is rather premature.

We thank referee for his/her supportive comment and hope that he/she will reconsider the maturity of our study after considering the new evidence in the revised manuscript.

References

Birbrair A, Zhang T, Wang ZM, Messi ML, Enikolopov GN, Mintz A, Delbono O (2013) Role of pericytes in skeletal muscle regeneration and fat accumulation. *Stem Cells Dev* 22: 2298-314

Dong Y, Silva KA, Dong Y, Zhang L (2014) Glucocorticoids increase adipocytes in muscle by affecting IL-4 regulated FAP activity. *FASEB J* 28: 4123-32

Ermolova NV, Martinez L, Vetrone SA, Jordan MC, Roos KP, Sweeney HL, Spencer MJ (2014) Long-term administration of the TNF blocking drug Remicade (cV1q) to mdx mice reduces skeletal and cardiac muscle fibrosis, but negatively impacts cardiac function. *Neuromuscul Disord* 24: 583-95

Grounds MD, Torrissi J (2004) Anti-TNFalpha (Remicade) therapy protects dystrophic skeletal muscle from necrosis. *FASEB J* 18: 676-82

Heredia JE, Mukundan L, Chen FM, Mueller AA, Deo RC, Locksley RM, Rando TA, Chawla A (2013) Type 2 innate signals stimulate fibro/adipogenic progenitors to facilitate muscle regeneration. *Cell* 153: 376-88

Kawanishi N, Mizokami T, Niihara H, Yada K, Suzuki K (2016) Macrophage depletion by clodronate liposome attenuates muscle injury and inflammation following exhaustive exercise. *Biochem Biophys Rep* 5: 146-151

Kitzmann M, Bonnieu A, Duret C, Vernus B, Barro M, Laoudj-Chenivresse D, Verdi JM, Carnac G (2006) Inhibition of Notch signaling induces myotube hypertrophy by recruiting a subpopulation of reserve cells. *J Cell Physiol* 208: 538-48

Kopinke D, Roberson EC, Reiter JF (2017) Ciliary Hedgehog Signaling Restricts Injury-Induced Adipogenesis. *Cell* 170: 340-351 e12

Kuru S, Inukai A, Kato T, Liang Y, Kimura S, Sobue G (2003) Expression of tumor necrosis factor-alpha in regenerating muscle fibers in inflammatory and non-inflammatory myopathies. *Acta Neuropathol* 105: 217-24

Madaro L, Passafaro M, Sala D, Etxaniz U, Lugarini F, Proietti D, Alfonsi MV, Nicoletti C, Gatto S, De Bardi M, Rojas-Garcia R, Giordani L, Marinelli S, Pagliarini V, Sette C, Sacco A, Puri PL (2018) Denervation-activated STAT3-IL-6 signalling in fibro-adipogenic progenitors promotes myofibres atrophy and fibrosis. *Nat Cell Biol* 20: 917-927

Mann CJ, Perdiguero E, Kharraz Y, Aguilar S, Pessina P, Serrano AL, Munoz-Canoves P (2011) Aberrant repair and fibrosis development in skeletal muscle. *Skelet Muscle* 1: 21

Mozzetta C, Consalvi S, Saccone V, Tierney M, Diamantini A, Mitchell KJ, Marazzi G, Borsellino G, Battistini L, Sassoon D, Sacco A, Puri PL (2013) Fibroadipogenic progenitors mediate the ability of HDAC inhibitors to promote regeneration in dystrophic muscles of young, but not old Mdx mice. *EMBO Mol Med* 5: 626-39

Mu X, Agarwal R, March D, Rothenberg A, Voigt C, Tebbets J, Huard J, Weiss K (2016) Notch Signaling Mediates Skeletal Muscle Atrophy in Cancer Cachexia Caused by Osteosarcoma. *Sarcoma* 2016: 3758162

Pannerec A, Formicola L, Besson V, Marazzi G, Sassoon DA (2013) Defining skeletal muscle resident progenitors and their cell fate potentials. *Development* 140: 2879-91

Radley HG, Davies MJ, Grounds MD (2008) Reduced muscle necrosis and long-term benefits in dystrophic mdx mice after cV1q (blockade of TNF) treatment. *Neuromuscul Disord* 18: 227-38

Rosen ED, MacDougald OA (2006) Adipocyte differentiation from the inside out. *Nat Rev Mol Cell Biol* 7: 885-96

Trask OJ, Jr. (2004) Nuclear Factor Kappa B (NF-kappaB) Translocation Assay Development and Validation for High Content Screening. In Assay Guidance Manual, Sittampalam GS, Coussens NP, Brimacombe K, Grossman A, Arkin M, Auld D, Austin C, Baell J, Bejcek B, Caaveiro JMM, Chung TDY, Dahlin JL, Devanaryan V, Foley TL, Glicksman M, Hall MD, Haas JV, Inglese J, Iversen PW, Kahl SD et al. (eds) Bethesda (MD):

Uezumi A, Ito T, Morikawa D, Shimizu N, Yoneda T, Segawa M, Yamaguchi M, Ogawa R, Matev MM, Miyagoe-Suzuki Y, Takeda S, Tsujikawa K, Tsuchida K, Yamamoto H, Fukada S (2011) Fibrosis and adipogenesis originate from a common mesenchymal progenitor in skeletal muscle. *J Cell Sci* 124: 3654-64

Thank you for transferring your revised manuscript entitled "Fibro-adipogenic progenitors of dystrophic mice are insensitive to NOTCH regulation of adipogenesis" to Life Science Alliance. Your manuscript was reviewed at another journal before and the editors transferred those reports to us with your permission.

The reviewers who evaluated your work at the other journal thought that the proteomic data and the potential role of Notch signalling in regulating adipogenic differentiation of FAPs is interesting. They would have expected further in vivo evidence for your conclusions. Lack of this additional support does not preclude publication here, and we would thus like to invite you to submit a final version of your work for publication in Life Science Alliance.

The remaining reviewer concerns should get addressed in a point-by-point response and via text changes to tone-down the conclusions/leave room for alternative explanations throughout the manuscript. Point 2 of referee #1 concerning the age of the mice used for the different analyses should get addressed in the text as well. Please also include the following slight changes to match LSA's formatting style / editorial requests:

- Please enter authors contributions and a summary blurb in our submission system
- The following callouts to figures are missing in the text, please add: 3D, 6D-F
- We display S figures in-line in the HTML version of the paper; please upload the S figures (currently in appendix file) as individual files and include the S figure legends in the main manuscript file.
- the legend for Fig 8 lists panels A-A-B, please correct to A-B-C
- Please include accession codes/identifies for the proteomics raw data and RNA-seq data

Referee 1:

Summary:

The revised manuscript includes additional experimentation intended to address concerns raised regarding dystrophic FAPs becoming insensitive to Notch regulation of adipogenesis. The proposed model of Notch antagonism of the adipogenic differentiation of FAPs is interesting and the potential for the coupling of inflammation/TNF/NFκB and Notch signaling is exciting. However, even with the additional experimentation, the results do not sufficiently support the authors' conclusions. There is little *in vivo* evidence to show that Notch acts on FAPs to inhibit adipogenesis. There can be multiple interpretations for the *in vivo* and *in vitro* experiments. Many of the new, *in vivo* experiments are redundant with previously published reports. The *in vitro* experiments are uncoupled by age with the *in vivo* experiments, leading to questions about their physiological relevance.

We do not understand this criticism. All our experiments either *in vivo* or *ex vivo* are performed with 6 weeks old mice.

In addition, the new in vitro experiments showing that Notch promotes FAP differentiation into myofibroblasts also support an alternative interpretation of the data (e.g. Notch promotes bipotential FAPs acquiring a fibrogenic cell fate rather than adipogenic cell fate).

The hypothesis made by the referee is subtly different from our simple interpretation of the results (adipogenesis of mdx FAPs is not sensitive to Notch inhibition). If I understand it correctly, he/she suggests that our preparation of FAPs contains cells that have the potential to differentiate into either adipocytes or fibroblasts and that in wt FAPs (not mdx) Notch favors the population with propensity for fibrogenesis. However, we have purified from both models a mononuclear population of cells that, although heterogeneous in some respects (see cytof data), share the property of being able to “homogeneously” differentiate into either adipocytes or fibroblasts. When we add Notch ligand, only FAPs from the wt model do not differentiate into adipocytes but rather increase their propensity to differentiate into fibroblasts. This experiment however was only reported in the “rebuttal” and it is not included in the manuscript. Although the hypothesis that emerges from our work is simpler and more consistent with the data, we have revised the text including the possibility of alternative explanations.

Response to the authors rebuttal:

Major Concerns:

1. Previous comment: Figures 1 and S2 use single cell proteomic analysis to show that FAPs derived from wild type, cardiotoxin-injured and mdx muscles differentially express surface antigens, particularly CD34 and Sca1. However, it is not entirely clear how these findings relate to the rest of experimentation in this manuscript. FAPs from WT, CTX and mdx sources all differentiate into adipocytes. There was no testing whether there is a difference in the adipogenic potential of the different subpopulations (CD34^{Hi}Sca1^{Hi} and CD34^{Lo}Sca1^{Lo}) of FAPs. Therefore, it seems that this experiment is unrelated. Experiments test the effects of FAPs source (WT, CTX, mdx) on adipogenic differentiation and NOTCH mediated inhibition of FAPs adipogenesis, but without a connection to the differential expression of the surface markers from FAPs of different origins.

- Authors' response: We understand the point made by the referee and we partly agree with him/her. There is no mechanistic connection between the observation of surface antigen expression in different FAP populations and their sensitivity to NOTCH ligand inhibition. However, we use the results of mass cytometry characterization to support a cell autonomous mechanism underlying the higher propensity of mdx FAPs to generate adipocytes. In principle

mdx FAPs could escape differentiation control either because of a defect in the environmental signals limiting adipogenesis; or because of a defect in the molecular machinery that senses these signals. In our first figure, we show that FAPs from mdx mice are in a different cell state when compared to wild type and, as such, may respond differently to environmental cues. In addition, the Cytof results represent a detailed characterization of the cell model that we are using in our studies, which has not been reported before. A PhD student in the group is now addressing functional heterogeneity in CD34+iSca1+ and has, for instance, shown that Sca1H are more adipogenic while SCA1L are more fibrogenic ex vivo. However, we feel that this characterization, albeit thought provoking, deviates from the main message of our report while opening new interesting avenues. If the referee thinks that this figure, as it is, is confusing without offering support to our conclusions we could drop it.

- Reviewer's response: It remains unclear how the CYTOF experiments fit the "main message" without evaluating the heterogeneous populations for differences in Notch receptor expression or functional ability to acquire an adipogenic phenotype.

We have discussed this point with the editor and concluded to leave the Cytof data.

2. Previous comment: Lack of in vivo support. The majority of the experimentation in this manuscript is in vitro. Although, the in vitro experiments appear to be thorough and well-designed, it is not clear if the in vitro findings translate to the in vivo system. For example, FAPS are cultured for extended periods, an attachment period followed by up to 8-days of stimulation, but it's not clear if these conditions are physiologically relevant without in vivo corroboration.

- Authors' response: Although we understand the point made by the referee, the adipogenic differentiation assay requires at least a week. The FAP differentiation protocol, as described in our manuscript (or similar), has been used in several reports (Dong, Silva et al., 2014, Mozzetta, Consalvi et al., 2013). What we show is that the FAPs from the two mouse models, wt and mdx, behave differently in in vitro assays (Figure 5 in the manuscript). We find it unlikely that this result is an artifactual consequence of the assay time. In addition, as suggested, in the revised manuscript we have added a new in vivo experiment supporting the cross talk between NOTCH and inflammatory signals in the control of adipogenesis.

- Reviewer's response: It is not clear how the in vitro experiments fit in vivo. Can the authors please clarify why fat deposition in vivo was demonstrated in mice older than 12 months of age but then FAPs isolated from 6-week old mdx mice were used for in vitro experiments?

As already stated, our experiments have been all performed with 6 weeks old mice. The confusion could arise from the fact that we characterize Notch insensitivity in young mdx mice, while fat deposition is observed in old mdx mice. However, we have dwelt at length in the discussion on how the interaction between inflammation and Notch explains this. We have carefully revised this session in the discussion aiming at improving clarity in order to avoid misunderstandings.

3. Previous comment: The greatest in vivo support for NOTCH inhibiting the adipogenic differentiation of FAPs is in Fig. 3K, which shows photomicrographs of muscle sections from CTX injured muscle also treated with NOTCH inhibitor DAPT i.p. which increased perilipin and sudan black labeling. Systemic administration of DAPT will also inhibit NOTCH signaling in other cells comprising muscle, including muscle stem cells. Conboy IM et al. 2003 previously demonstrated that the administration of a Notch inhibitor impairs muscle regeneration and promotes tissue degeneration. Therefore, there is little support of an in vivo effect of Notch signaling being important regulator of FAP differentiation.

- Authors' response: We agree that i.p. administration of DAPT will affect the whole stem cell niche. In principle other cell types could differentiate into adipocytes and cause fat deposition. However, FAPs are the major source of fat infiltrations in the skeletal muscle tissue (Uezumi, Ito et al., 2011). In our experiment, we also checked the effect of the treatment on myofiber regeneration (Fig. 1). The myofiber cross-sectional area analysis shows a slight increase of the cross sectional area in the DAPT treated group compared to control, but also a higher number of centrally nucleated myofibers, indicating that the regeneration process is still active after 21 days. These two altered readouts are most likely the consequence of DAPT treatment on satellite activation and proliferation. These observations seem to be in contrast, at least in part, with the results of Conboy et al.

However, it must be emphasized that the experimental conditions are considerably different. Conboy et al 2003, by measuring the fraction of center nucleated fibers, conclude that Notch inhibition in vivo negatively affects muscle regeneration. We designed our experiments to minimize the effect of Notch inhibition on myofiber regeneration. Conboy et al injected the NOTCH inhibitor 48h after injury and analyzed regeneration at an early time, 3 days later. As an additional difference, they did not damage the muscle by ctx injury but rather by piercing the muscle multiple times with a needle. Finally, they did not look at other markers of regeneration/degeneration (eg, CSA, fat infiltration, fibrotic infiltrations). Since activation and differentiation of satellite cells requires Notch activity in several phases of the regeneration process, in order to minimize interference with the activation and proliferation phase, we administrated DAPT three days after cardiotoxin injury, and we analyzed our readouts 21 days post-injury, when the muscle architecture is restored. Because of these experimental differences, it is therefore difficult to compare the results of the two experiments. Finally, our observation of a slightly higher CSA is consistent with reports outlining that Notch inhibition causes hypertrophy (Kitzmann, Bonniou et al., 2006, Mu, Agarwal et al., 2016).

Concerning in vivo experiment, (DAPT ip injection) we repeated DAPT injection increasing the total number of animals to n=4 for each group (control and DAPT) and quantifying the perilipin positive fat infiltration area. We have now included these results in panel k of Fig.4.

- Reviewer's response: Thank you for the explanation.

OK

4. Previous comment: An experiment that included the conditional overexpression of notch intracellular domain (NICD) in FAPs using a PDGFRA-Cre system to show reduced fatty tissue infiltration in the muscle would greatly enhance the impact of this experimentation.

- Authors' response: As a general consideration, all the experiments suggested by the referees required the sacrifice of animals, either for in vivo procedures or for purification of primary cells. Our protocols for this project, as approved by the University ethical committee and by the Health Ministry, allows for the use of a limited number of wild type or mdx genetic background animals. Extending the number of wild type or mdx animals to be treated with the approved protocols is possible with a relatively simple procedure. However, in order to perform the experiment suggested by the referee it is necessary to introduce in the project new genetic backgrounds and/or new procedures, that had not been included in our original applications to the animal ethic committee. Thus, we would need a qualitative and quantitative extension, which, with the current Italian regulation, needs to be approved at the national level. In addition, to our knowledge, PDGFRA-Cre and a lox mouse expressing NICD under the control of Cre cannot be easily purchased/obtained, hence, it would require constructing new transgenic mice, which would lead us beyond the time frame that we have set for this project. Moreover, if we understand correctly the experiment proposed by the referee, he/she suggests to "cure" the FAP mdx insensitivity to NOTCH ligands by forcing the expression of NCID. This would require the engineering of a mouse conditionally expressing NCID in an mdx background with considerable extra breeding/crossing/testing. Finally, our in vivo readout requires growing the animals for at least a year when fat infiltrations can be observed in mdx mice. The FAPs that we use in our ex vivo experiments are purified from 6 weeks old mice when FAPs are already insensitive to NOTCH inhibition but no fat is observed because of the compensating effect of the inflammatory environment (Figure 7 in manuscript).

- Reviewer's response: Minimizing the use for additional animals is an important consideration, however additional experimentation is needed to confirm findings in vivo under physiological conditions. There are commercially available mouse lines to manipulate Notch signaling through Jackson labs PDGFRA-Cre #013148 and N1-IC #008159 as used in Mourikis et al 2012 (PMID 22069237). In the authors response to the review they outline an experiment that would greatly enhance the strength of their findings. There are also other approaches that could be used to test the importance of Notch signaling as a switch for adipogenesis in FAPs in vivo. For example, using a commercially available conditional mutant to show that genetic inhibition of Notch promotes adipogenesis after acute muscle injury would have been more compelling.

See response to the editor.

Experimentation using clodronate mediated macrophage depletion is largely redundant of previously published findings. Several groups have shown that depleting leukocytes (CD11b-DTR mice) or impairing monocyte/macrophage recruitment into the muscle (CCR2^{-/-} mice)

promotes muscle degeneration and fat deposition (Martinez et al. 2010 PMID: 20631294). Additionally, it has been shown that both impairing macrophages (CCR2^{-/-} mice), conditional ablation of TNF from myeloid cells (LysM-Cre/TNF flox mice) and anti-TNF treatment promotes the survival FAPs (Lemos DR et al. 2015 PMID: 26053624). Although that investigation did not assay for changes in adipogenesis, they did find that macrophages acquired a noncanonical phenotype that promotes the survival of FAPs on a similar time course that was used in this investigation and could represent an alternative mechanism for the in vivo data to the one proposed by the authors.

Our experiments were aimed at demonstrating the synergy between inflammatory signals and Notch inhibition in vivo. We used clodronate treatment as a well-characterized tool to deplete macrophage and we assayed fat deposition after macrophage depletion and Notch inhibition as single treatments and in combination. We thank the referee for pointing out the reports describing muscle regeneration in mouse models that are defective in macrophage recruitments (we have now referred to these in our manuscript) but we think that none of these addresses the crosstalk between TNF α and NOTCH inhibition on fat deposition during muscle regeneration. Concerning the negative effect of TNF α on FAP survival, we have not been able to reproduce the observation of Lemos et al. In our conditions, TNF α , even at high concentrations does not promote apoptosis of FAPs and, as admitted by the referee this report does not address the effect of these treatments on adipogenesis.

Minor Concerns:

5. Previous comment: How long is the 'attachment period' between the FAPs isolation and cell stimulation'? Is this normalized between all preps? Do FAPs from WT, CTX or mdx attach differently?

- Authors' response: We apologize for not providing this information. The attachment period in all experiments was 4 days. As far as we can judge, FAPs from wild type and mdx mice attach equally to plastic although this is difficult to evaluate with precision because the FAPs from mice of the two different genetic backgrounds have a different proliferation rate in the "attachment period". However, we quantified and compared the number of wt and mdx FAPs at day 0, which is the 4th day after seeding, and at day 8. We did not observe significant difference in cell number. In addition, we measured the percentage of proliferating cells with EdU proliferation assay at day 0 and did not observe significant difference between wt and mdx FAPs (Fig.2).

- Reviewer's response: Thank you for the additional experimental details.

OK

6. Previous comment: Results, subsection 2, 1st paragraph, last sentence. "Cell proliferation" and "survival" were not assayed. Nuclei/field was assayed, which would reflect differences in cell

accumulation. Although no change in cell accumulation likely means no change in cell proliferation or survival, the conclusion is not proven definitively.

- Authors' response: We thank the referee for pointing this out. We now rephrased the text "...without affecting cell number".

OK

7. Previous comment: Results, subsection 2, 2nd paragraph, last sentence. 'We conclude that FAPs, by synthesizing both a Notch ligand and receptor are able to limit their own differentiation by an autocrine mechanism.' There is limited experimental support of this. These conclusions are based on a difference in adipogenic differentiation in the presence of Notch activators and inhibitors. The authors did not look at Notch receptor or ligand expression in FAPs.

- Authors' response: We thank the referee for his/her suggestion. We now evaluated the expression of the members of Notch signaling pathway by western blotting (Fig. 3). Since the expression level of these membrane proteins is relatively low, we could not detect them in the proteomic experiments. However, we have now performed a new RNAseq experiment to characterize the transcriptome of wt and mdx FAPs. These data are now included as supplementary materials. As shown below, our data show that FAPs express both NOTCH ligands and receptors and have therefore the potential to autoregulate their differentiation via an autocrine mechanism.

- Reviewer's response: The additional RNA data strengthen the authors' conclusion. Did the authors consider immunocytochemistry or FACS for Notch receptors on cells if low abundance of protein is an issue?

We did not have suitable antibodies for immunocytochemistry of NOTCH. On the other hand, we believe that the combination of western blotting and RNAseq data supports the potential of FAPs to regulate their adipogenesis via an autocrine mechanism mediated by the Notch pathway.

- Authors' response: The main question of this study was the effect of Notch signaling on adipogenic differentiation of FAPs. Nonetheless, we did analyze fibrogenic differentiation of FAPs upon activation/inhibition of the Notch signaling pathway (Fig. 4) and we observed an increase in aortic smooth muscle actin (SMA) expression in the group treated with Notch ligand. SMA is a common marker of myofibroblasts, activated form of fibroblast responsible for transient production of extracellular matrix during tissue regeneration after acute injury (Mann, Perdiguero et al., 2011).

- Reviewer's response: These new data suggest the possibility that Notch signaling promotes the differentiation of FAPs into myofibroblasts (alpha-smooth muscle actin+ cells) rather than inhibiting adipogenesis per se. Can the authors address this alternative? Was fibrosis assessed in DAPT treated mice? Assays for collagen deposition? aSMA+ cells?

The mechanisms that underlie the differential response to Notch inhibition in the two model systems are not characterized in detail in our report. At a certain stage of their differentiation process FAPs have to decide whether to become an adipocyte or a fibroblast. What we observe experimentally is that in the presence of Notch they do not become adipocytes and a certain fraction start synthesizing SMA. We say that Notch inhibits adipogenesis. The referee prefers to say that Notch favors fibrogenesis. We feel that in essence we are saying the same thing.

- Authors' response: In order to evaluate the effect of the WNT and Hedgehog pathway perturbations on fibrogenesis we would require the sacrifice of more animals. Since WNT and HH modulation of FAP differentiation was not the central part of the current work we decided not ask the animal committee approval for extension of the protocol.
- Reviewer's response: OK.

OK

8. Previous comment: Results, subsection 4, 2nd paragraph. Inconsistent reference style. Grounds et al. and Lemos et al. are not numerically annotated.

- Authors' response: We thank the referee for pointing this out. We corrected it.
- Reviewer's response: OK.

OK

9. Previous comment: The authors do a nice job of showing that TNF when synergistically combined with NOTCH signaling inhibits mdx FAPs differentiation. Also, I agree that the source of TNF is likely to be inflammatory cells. Yet, I would like to know if FAPs from WT, CTX or mdx differentially express TNF and could regulate adipogenic differentiation in an autocrine or paracrine manner.

- Authors' response: To respond to the referee's comment, we analyzed TNF α expression in FAPs from wt, mdx and ctx treated muscles by single cell mass cytometry, and identified a small subpopulation which is positive for TNF α and pSTAT1 (Fig. 5 and 6). Interestingly, this population was more abundant in the FAPs purified from animals treated with cardiotoxin three days post injury. Three days post injury corresponds to the peak of TNF α expression during muscle regeneration (Lemos et al. 2015). However, given the small size of the FAP subpopulation expressing TNF α , it is unlikely that this represents a major contribution to TNF α levels in the skeletal muscle. In Fig. 7 is reported an experiment where we treated mdx FAPs seeded on DLL1 in growth medium with anti-TNF α antibodies and we observed a slight increase in adipogenic differentiation, consistent with autocrine regulation mediated by TNF α . However, this difference was not statistically significant.

• Reviewer's response: Thank you for addressing this comment with additional experimentation. These are interesting data and the authors interpretation is reasonable. Did the authors confirm that clodronate macrophage depletions reduced TNF in whole muscle?

No, we did not measure TNFa in clodronate treated muscles.

Referee 2:

The authors have been partially responsive to the first review. They provided new data for some of the points raised in my previous review, and these data are supportive of their claims. However, I still feel that the paper is unfocused and some parts are confusing and I was disappointed that the authors were not willing to perform in vivo studies also requested by Reviewer 1. Issues of animal ethics committees are irrelevant to scientific merit, so should not be discussed in rebuttals. This detracts from my initial enthusiasm for the study, therefore I will give it a weak pass.

Please see response to the editor and previous rebuttal for ethical issues.

Referee 3:

As this is a re-review, I will not re-summarize the findings in this study but upon the major points of the revision. Overall, I find the authors have made a big effort to address the concerns, and with regard to my specific suggestion of not relying upon a single pharmacological approach, have added RNA interference assays that strengthen the conclusions. I feel, as to the other reviewers, that the in vivo aspect of the study is still a bit weak, and that the 'excuse' of re-justifying more animals is a hardship is not really an adequate reason for not addressing this further, nonetheless, the central point(s) of this study are convincing.

OK

I should point out to the authors that while the original studies showing FAPs as the major source of fat were suggestive of this conclusion, they were by no means in vivo proof. That has finally come from a more recent study (Stumm et al., 2018 Odd skipped-related 1 (Osr1) identifies muscle-interstitial fibro-adipogenic progenitors (FAPs) activated by acute injury) in which this is shown through lineage tracing in vivo. While this result was expected, it is important to note that the in vivo evidence has been supplied.

We thank the reviewer for pointing this out. We had missed the relevance of this report. We have now added a brief comment on this work.

I am also a bit confused about the poor survivability of FAPs following injection into muscle as claimed by the authors. This has been done by several groups.

We are aware that several groups have done transplantation of FAPs into muscles; however, we were not as successful. In order to perform the experiment we would require more time for the optimization of the protocol that would also lead to significant increase in the number of animals. Considering the time given for the revisions and the uncertainty of the successful experimental outcome, we have decided not to pursue this strategy.

Despite these negative comments, the high quality of the proteomic work and the uncovering of the NOTCH pathway as a player in driving cell fate is of high interest and merits publication.

We are pleased to see that we have convinced this reviewer that our work, despite some limitations, merits publication.

June 7, 2019

RE: Life Science Alliance Manuscript #LSA-2019-00437-TR

Prof. Gianni Cesareni
University of Rome Tor Vergata
Department of Biology
Via della Ricerca Scientifica
Rome, italia 133
Italy

Dear Dr. Cesareni,

Thank you for submitting your Research Article entitled "Fibro-adipogenic progenitors of dystrophic mice are insensitive to NOTCH regulation of adipogenesis" to Life Science Alliance. I appreciate your final response to the reviewers and the introduced changes and it is a pleasure to let you know that your manuscript is now accepted for publication in Life Science Alliance. Congratulations on this interesting work.

*****IMPORTANT:** If you will be unreachable at any time, please provide us with the email address of an alternate author. Failure to respond to routine queries may lead to unavoidable delays in publication.*******

DISTRIBUTION OF MATERIALS:

Again, congratulations on a very nice paper. I hope you found the review process to be constructive and are pleased with how the manuscript was handled editorially. We look forward to future exciting

submissions from your lab.

Sincerely,
